# Folliculin directs the formation of a Rab34–RILP complex to control the nutrient-dependent dynamic distribution of lysosomes

Georgina P Starling[1,†], Yan Y Yip[1,†], Anneri Sanger[1], Penny E Morton[1], Emily R Eden[2] & Mark P Dodding[1,*]

## Abstract

The spatial distribution of lysosomes is important for their function and is, in part, controlled by cellular nutrient status. Here, we show that the lysosome associated Birt–Hoge–Dubé (BHD) syndrome renal tumour suppressor folliculin (FLCN) regulates this process. FLCN promotes the peri-nuclear clustering of lysosomes following serum and amino acid withdrawal and is supported by the predominantly Golgi-associated small GTPase Rab34. Rab34-positive peri-nuclear membranes contact lysosomes and cause a reduction in lysosome motility and knockdown of FLCN inhibits Rab34-induced peri-nuclear lysosome clustering. FLCN interacts directly via its C-terminal DENN domain with the Rab34 effector RILP. Using purified recombinant proteins, we show that the FLCN-DENN domain does not act as a GEF for Rab34, but rather, loads active Rab34 onto RILP. We propose a model whereby starvation-induced FLCN association with lysosomes drives the formation of contact sites between lysosomes and Rab34-positive peri-nuclear membranes that restrict lysosome motility and thus promote their retention in this region of the cell.

**Keywords** BHD syndrome; folliculin; lysosome; Rab34; RILP
**Subject Categories** Membrane & Intracellular Transport; Metabolism

## Introduction

Recent evidence indicates that the dynamic spatial positioning of lysosomes within the cytoplasm plays an important role in several of their functions, including as the terminal degradative compartment of the autophagic pathway, the site of amino acid and growth factor signal integration to the mTORC1 kinase, and in their role as a secretory organelle [1–4]. Lysosomes are transported, bi-directionally, on the microtubule network by cytoplasmic dynein

1 and several kinesin family motors [5–12]. Several of these motors are recruited by lysosome associated small GTPases and their effector proteins [1]. A number of pathways leading to motor recruitment and modulation of lysosome transport/distribution have emerged. To drive transport towards the plus end of microtubules that are typically located at the cell periphery, the small GTPase Arl8 recruits SKIP via its RUN domain, which in turn directly interacts with kinesin-1 [8,13]. The recently described BORC complex initiates this process by recruiting Arl8 [4]. Kinesin-1 can also be recruited by Rab7 though its effector FYCO1, which is loaded onto late endosomes at ER contact sites [14,15]. To promote transport towards the minus end of microtubules that are predominantly located in a peri-nuclear position at the microtubule organising centre, Rab7 can also interact with RILP which in turn recruits cytoplasmic dynein 1 via the p150[glued] subunit of dynactin [16–18]. Taken together, these studies imply that summation of various opposing centripetal and centrifugal directed activities help to define the steady state distribution of this highly dynamic organelle. Within many cell types, this is manifested as a cluster of peri-nuclear lysosomes with more dispersed lysosomes distributed throughout the cytoplasm and at the cell periphery, with dynamic exchange between these populations [5].

In addition to these lysosome intrinsic components, other reports have shown that the lysosome extrinsic, predominantly Golgi localised, Rab34 and Rab36 GTPases can also impact upon lysosome distribution by promoting their peri-nuclear clustering [19–22]. For Rab34, this requires interaction with its effector RILP, which it also shares with Rab7. Consistent with the notion that Rab34 can impact lysosome distribution from a non-lysosomal membrane compartment, removal of its C-terminal lipidation sequences and substitution with Golgi targeting transmembrane domains did not impair its ability to induce peri-nuclear clustering [19].

Lysosome distribution is regulated by cellular nutrient status [2]. In HeLa cells, withdrawal of serum and amino acids (starvation), which suppresses mTORC1 activity, also promotes peri-nuclear clustering of lysosomes. Relative dispersion and accumulation at the cell periphery occurs in nutrient replete, high mTORC1 activity

1 Randall Division of Cell and Molecular Biophysics, King's College London, London, UK
2 Institute of Ophthalmology, University College London, London, UK
*Corresponding author. Tel: +44 207 848 6473; E-mail: mark.dodding@kcl.ac.uk
†These authors contributed equally to this work

conditions. Inhibition of peri-nuclear clustering by kinesin/Arl8b over-expression reduced autophagic flux and lysosome positioning regulates recovery of mTORC1 activity after starvation. The mTOR kinase is activated on the lysosomal surface, via a signalling network composed of Rag GTPases, the vacuolar ATPase, Ragulator complex, Gator complex and the folliculin (FLCN)/FNIP complex. This platform senses lysosomal amino acid levels and integrates other signalling inputs from energy, oxygen and growth factors [23,24]. Relatively little is known about how the nutrient signalling machinery co-ordinates with the transport machinery which controls the dynamic distribution of the lysosomal membrane compartment.

Disruption of *FLCN* causes the inherited kidney cancer disorder, Birt–Hoge–Dubé (BHD) syndrome [25–27]. The *FLCN* gene encodes a protein of 64 kDa that contains an N-terminal Longin domain and C-terminal DENN domain and lacks primary sequence homology to other mammalian proteins [28]. FLCN forms a complex with two other proteins FNIP1 and FNIP2, that also contain DENN and Longin domains, that can homo and heterodimerise, and are homologues of the *S. cerevisiae* protein Lst4 [29,30]. The N-terminal Longin region of FLCN shares homology with yeast Lst7 which forms a complex with Lst4, is encoded by a gene originally identified in a screen for synthetic lethality with the COPII component Sec13 and plays a crucial role in the amino acid-dependent trafficking of the *S. cerevisiae* amino acid permease GAP1p to the plasma membrane [31,32]. Lst7 lacks the C-terminal DENN domain found in FLCN.

The FLCN/FNIP complex receives signalling inputs from metabolic pathways as it is phosphorylated downstream of activation of mTORC1 and AMPK [33–36]. FLCN/FNIP associates with lysosome following serum and amino acid withdrawal, binds nucleotide free RagA/B and acts as a GTPase activating protein (GAP) for RagC to promote the recruitment and activation of mTORC1 on lysosomes [37–39], although FLCN loss in BHD syndrome can result in elevated mTORC1 activity in kidney tumours [40,41]. The orthologous Lst7–Lst4 complex in yeast functions in a similar manner [29,42]. Reports also suggest that FLCN/FNIP play a role in a range of other often ostensibly mechanistically distinct cellular processes. FLCN/FNIP loss impacts upon on cell migration/adhesion [43,44], TGF-β signalling [45,46], HIF1-α transcription [47], autophagy [48,49], ciliogenesis [50] and, via mTORC1 and TFEB/TFE3, regulates lysosome biogenesis and exit of stem cells from pluripotency [37,39,51,52] and several others, reviewed in [53]. Thus, a major

challenge for the field has been to integrate often quite disparate phenotypic and mechanistic data and to determine a coherent molecular mechanism for the action of FLCN.

The recent definition of the FLCN/FNIP complex as a lysosome associated multi-DENN, multi-Longin domain assembly prompted us to hypothesise that FLCN may regulate membrane traffic. Here, we present evidence consistent with that proposition, demonstrating that FLCN promotes the starvation- and Rab34-dependent redistribution of lysosomes to the peri-nuclear region by promoting the association of Rab34 with its effector RILP. We suggest that that this may occur at novel membrane contact site.

# Results

## FLCN is required for starvation-induced peri-nuclear lysosome clustering

As recent reports have suggested that association of endogenous FLCN with lysosomes is enhanced by serum/amino acid withdrawal [37–39], we compared immunofluorescence staining for FLCN and the late endosomal(LE)/lysosomal marker LAMP1 in cells cultured in normal growth media (DMEM, 10% FCS) to cells starved for 4 h of serum and amino acids in Krebs-Ringer bicarbonate buffer solution. LAMP1 staining does not differentiate between LE and lysosomal compartments, but for ease of reading, we will refer to both as lysosomes. We confirmed two independently reported observations: firstly, relatively little FLCN was detected in association with lysosomes under normal growth conditions, but association was dramatically enhanced by starvation (Fig 1A and B). Secondly, starvation induced the peri-nuclear clustering of lysosomes (Fig 1A). As expected, this starvation protocol suppressed mTORC1 signalling as measured by levels of phosphorylated-S6K and 4EBP and also resulted in a slight increase in the electrophoretic mobility of FLCN that is thought to occur as a result of a change in its phosphorylation state (Fig 1B) [33]. To test whether this correlation of FLCN lysosome association and distribution results from a functional connection, we used siRNA to deplete FLCN from HeLa cells (Fig 1C). Cells were co-labelled with LAMP1 and the Golgi marker Giantin to highlight starvation-induced clustering of lysosomes in this region of the cell (Fig 1D). Depletion of FLCN did not strikingly affect lysosome distribution in normal growth conditions although

**Figure 1. FLCN is required for starvation-dependent peri-nuclear clustering of lysosomes.**

A   Widefield immunofluorescence images of PFA-fixed HeLa cells showing endogenous FLCN and LAMP1 staining in (top) normal growth or (bottom) starvation conditions. Orange arrows highlight association of FLCN with LAMP1-positive membranes in the peri-nuclear region. White dotted line shows cell periphery in starved condition. Scale bar, 10 μm.

B   Quantification of FLCN/LAMP1 association (left) in growth and starvation conditions. A cell is scored as positive if 5 or more discrete FLCN/LAMP1 puncta were observed at 100× magnification using a widefield microscope. Data represent 60 cells from 3 independent experiments, error bars show SEM, \*\*\*$P < 0.001$ (two-tailed *t*-test). Western blot (right) shows levels of phosphorylated S6K and 4EBP in whole-cell extracts under the same conditions.

C   Western blot showing typical efficiency of FLCN knockdown in HeLa cells using a pool consisting of 4 oligonucleotides and 2 independent oligonucleotides from the pool. Graph shows relative FLCN expression ($n = 3$, error bars show ± SEM).

D   Representative confocal fluorescence images showing LAMP1 (magenta) distribution in HeLa cells under normal growth or starvation conditions in cells depleted of FLCN or control cells transfected with a non-targeting siRNA. The location of the Golgi (green) is highlighted by giantin staining. Scale bar, 10 μm.

E   Schematic illustrating application of the cumulative intensity distribution method for quantification of lysosome distribution.

F   Graph showing cumulative distribution of LAMP1 intensity in normal growth and starvation conditions in HeLa cells. *P*-value is determined by the extra sum of F-squares test following nonlinear regression and curve fitting. Error bars show ± SEM from 30 cells in 3 replicates.

G   Graph showing cumulative distribution of LAMP1 intensity in starvation conditions for control cells transfected with non-targeting siRNA or cells depleted of FLCN. Error bars show ± SEM from 30 cells in 3 replicates. *P*-value is determined by the extra sum of F-squares test following nonlinear regression and curve fitting.

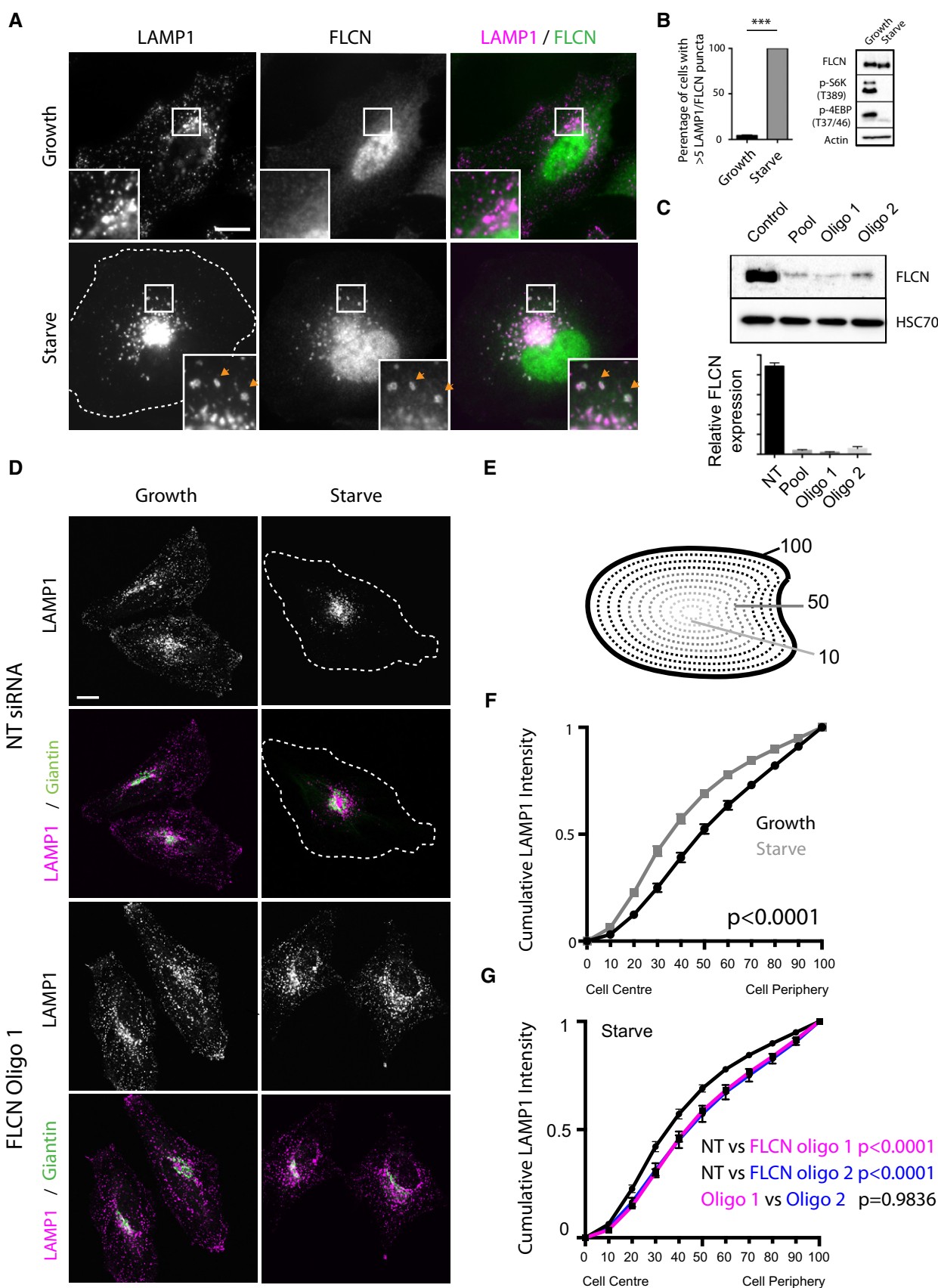

**Figure 1.**

peripheral accumulations of LAMP1 were more prominent (Fig 1D). Under starvation conditions, however, the peri-nuclear clustering of lysosomes was inhibited by FLCN depletion (Fig 1D), suggesting a functional connection between FLCN lysosome association and lysosome dynamics.

As the cytoplasmic distribution of lysosomes does vary from cell to cell [5,54] (see field in Fig EV1B), we sought to establish a robust, unbiased assay to assess these effects across a population of cells. To this end, images of cells were segmented by scaling the perimeter in 10% decrements. Cumulative integrated LAMP1 intensity (relative to the whole cell) was then plotted (Figs 1E–G and EV1A). Curve fitting using nonlinear regression allows statistical comparison of data sets using the extra sum of *F*-squares test. We validated this approach by examining changes in distribution by over-expressing GFP-RILP (which promotes the extreme peri-nuclear clustering and fusion of LE/lysosomes) or myc-SKIP (which promotes dramatic dispersion; Fig EV1A) [8,16,17]. By definition, such curves must start and finish at $x(0,0)$ and $y(100,1)$, respectively, deflection of the curve to the left indicates a shift to relatively more centralised (peri-nuclear) distribution, whereas a shift to the right indicates dispersion. We also validated this assay using an environmental manipulation that does not require protein over-expression by examining the acidification induced dispersal of lysosomes [54]. The level of response was less dramatic than with SKIP/RILP over-expression, but as expected, incubation of cells in acetate Ringer's buffer at pH 6.5 for 30 min resulted in a significantly more dispersed LAMP1 phenotype than in cells incubated in Ringer's buffer pH 7.4 (Fig EV1B), indicating that this assay is suitable for quantitatively describing changes in lysosome distribution induced by cell culture conditions. Application of this quantitative approach confirmed that FLCN depletion significantly suppresses starvation-induced lysosome clustering (Fig 1F and G). Given that FLCN association with lysosomes requires the FNIP proteins, we tested their role in this process. Transfection of cells with siRNA targeting both FNIP1 and FNIP2 did not affect lysosome distribution in normal growth conditions, but did inhibit starvation-induced clustering (Fig EV2A and B). Together, these data suggest that the FLCN/FNIP complex is required for nutrient-dependent control of lysosome distribution.

### FLCN/FNIP over-expression promotes the formation of dynamic lysosome associated tubules

Recent reports have demonstrated that FLCN-GFP associates with lysosomes when co-expressed with its FNIP binding partners [37,38]. We confirmed this by co-transfection of FLCN-GFP and HA-FNIP2 in HeLa cells and identified GFP-positive acidic organelles with Lysotracker-Red (Fig 2A and Movie EV1). Transfection with HA-FNIP2 had a similar effect on recruitment of endogenous FLCN to lysosomes (Fig EV2C). We obtained similar results in all experiments

for HA-FNIP1. Live-cell spinning disc confocal imaging of FLCN/FNIP2 transfected cells revealed three distinct classes of FLCN-GFP/Lysotracker-Red dynamics (Fig 2A). Peri-nuclear, FLCN-positive lysosomes tended to be slightly larger and engaged in short saltatory movements (Fig 2Ai and Movie EV1 zoom panels), whereas peripheral FLCN-positive lysosomes moved rapidly along linear trajectories with velocities over 1 µm/s (Fig 2Aii and Movie EV1 zoom panels). We also noted FLCN-GFP association with dynamic tubules, that co-labelled weakly or not at all with Lysotracker-Red, whose tips moved at speeds of up to 0.6 µm/s and emanated from larger peri-nuclear FLCN-GFP-positive compartments (Fig 2Aiii and Movie EV1 zoom panels, Fig 2B and Movie EV2). Fig 2Aiii shows a typical example of a FLCN-GFP tubule (usually 1–2 per cell). Fig 2B and Movie EV2 show a region of cell with a large number of such tubules. These tubules were disrupted and shortened by fixation (visible in approximately 5% of cells), but when successfully methanol fixed, were observed to emanate from LAMP1-positive structures and stained strongly for Rab7, consistent with a LE/lysosomal origin (Fig 2C). These structures were induced by FLCN/FNIP2 expression as we did not observe them in control cells stained for Rab7. Although we did not observe striking FLCN/FNIP2 induced peri-nuclear clustering in these conditions, morphologically and dynamically similar tubules have been observed in macrophages following LPS stimulation and require the activity of several lysosome associated small GTPases, effector and motor proteins [55] and so are consistent with our hypothesis that FLCN/FNIP have an intrinsic capacity to impact upon lysosome membrane dynamics [55].

### FLCN supports Rab34-induced peri-nuclear lysosome clustering

Given that the FLCN/FNIP complex is comprised of multiple Longin and DENN domains, we considered the small GTPases and their effector molecules that are known to play a role in promoting centripetal lysosome transport: Rab7, Rab34, Rab36 and their shared effector RILP. Consistent with [56], we were unable to detect Rab36 expression in HeLa and so we focused on Rab7 and Rab34. Under normal growth conditions, expression of GFP-Rab34 promoted the robust peri-nuclear accumulation of lysosomes compared to expression of GFP alone, whereas expression of GFP-Rab7 had no significant effect (Fig 3A). To ask whether Rab34 may be regulated by nutrient status, we performed a GST-RILP effector pull-down assay, taking advantage of the preferential interaction of RILP with the GTP bound, active, form of Rab34 [19]. Consistent with nutrient regulation, starvation resulted in a ≈ 2-fold increase in endogenous Rab34 retention on GST-RILP resin (Fig 3B). Next, we tested whether FLCN plays a role in lysosome clustering induced by GFP-Rab34. Depletion of FLCN in a population of cells matched for Rab34 expression level, using either of 2 siRNAs, significantly

**Figure 2.** **Distribution and dynamics of FLCN-GFP in HeLa cells.**

A  Representative spinning disc confocal image showing FLCN-GFP and lysosomes (Lysotracker-Red) in a live FLCN-GFP/HA-FNIP2 transfected HeLa cell (from Movie EV1). Top panels show the first frame of a 180-frame (1fps) image series. Bottom panels show a maximum intensity projection of the image series to highlight mobile and immobile components. Boxed and magnified regions show typical peri-nuclear (i) and peripheral (ii) regions in the cell as well as a FLCN-GFP tubule (iii). Scale bar, 5 µm.

B  Spinning disc confocal image series of a region live FLCN-GFP/HA-FNIP2 transfected HeLa cell highlighting dynamic FLCN-GFP tubules (Movie EV2). First merge panel shows FLCN-GFP and Lysotracker-Red at *T* = 0. Subsequent panels show only FLCN-GFP at 30-s intervals.

C  Confocal image of a region of a methanol-fixed FLCN-GFP/HA-FNIP2 transfected HeLa cell showing Rab7 (red) and LAMP1 (blue) association with FLCN-GFP tubules. Scale bar, 1 µm.

    

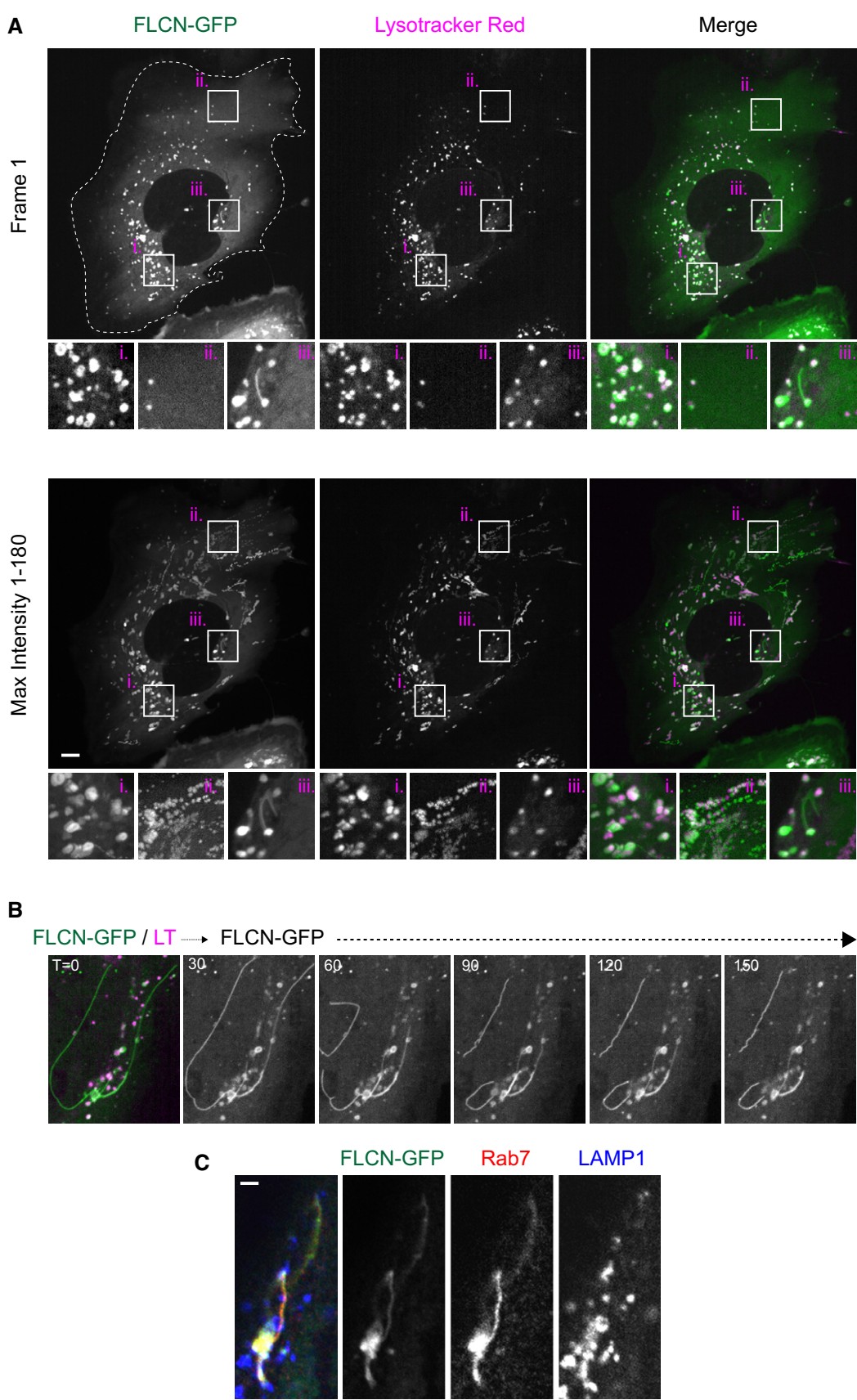

**Figure 2.**

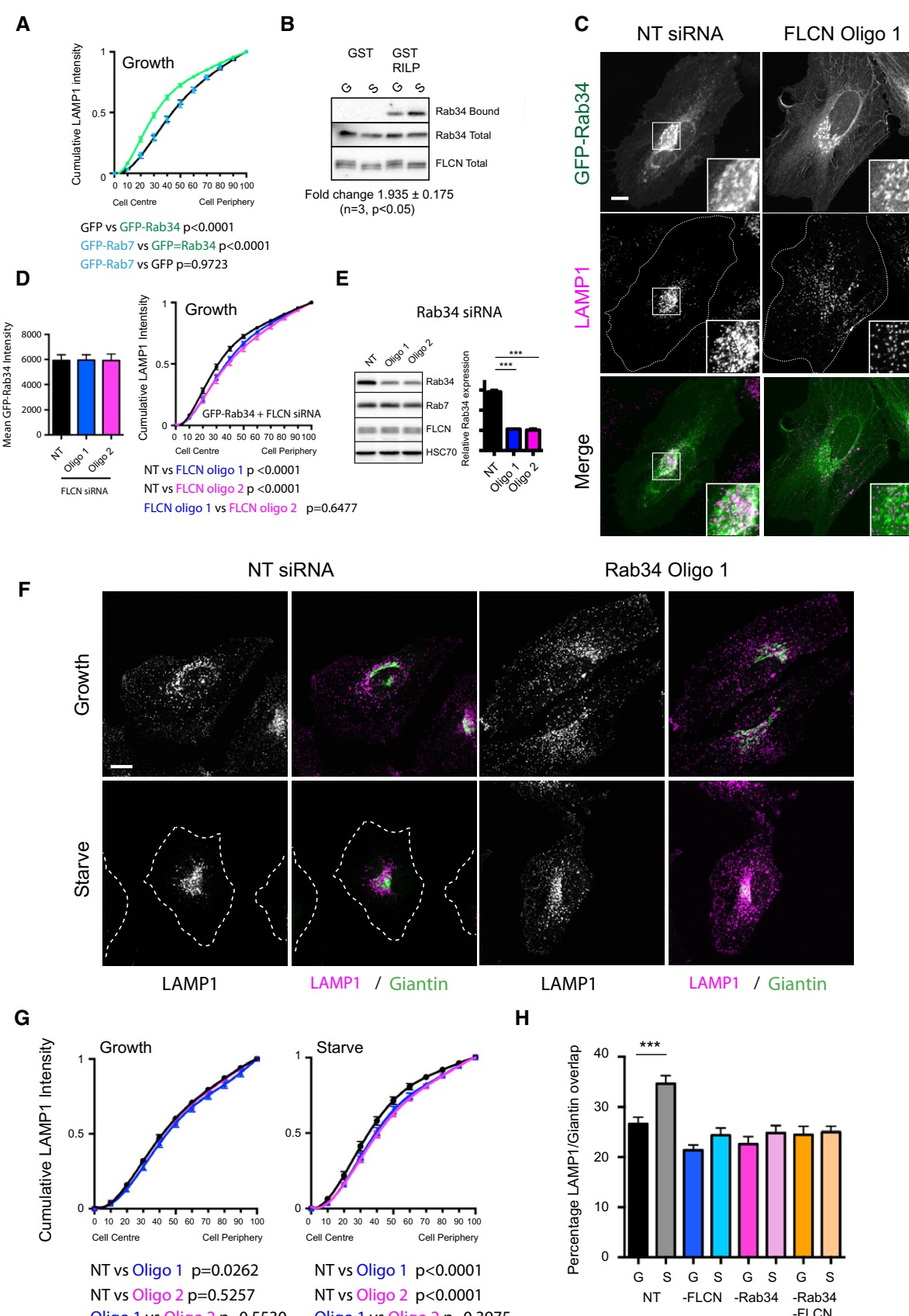

Figure 3.

**Figure 3.   FLCN supports Rab34-dependent peri-nuclear lysosome clustering.**

A    Graphs showing cumulative distribution of LAMP1 intensity in control GFP (black), GFP-Rab34 (green) or GFP-Rab7 (blue) transfected cells. Note black line and symbols partially obscure blue. *P*-value is determined by the extra sum of F-squares test following nonlinear regression and curve fitting. Error bars show ± SEM from 30 cells in 3 replicates.

B    GST-RILP pull-down analysis from HeLa cells in normal and starved conditions showing an enhanced capacity of endogenous Rab34 to bind RILP in starvation conditions. Quantification is mean of 3 independent experiments. Error is SEM (two-tailed *t*-test).

C    Representative confocal fluorescence images of FLCN depleted or control cells, transfected with GFP-Rab34. Scale bar, 10 μm.

D    Graphs showing mean GFP-Rab34 expression (left) and cumulative distribution of LAMP1 intensity (right) in FLCN depleted or control siRNA transfected HeLa cells expressing GFP-Rab34. Error bars show ± SEM from 30 cells in 3 replicates. *P*-value is determined by the extra sum of F-squares test following nonlinear regression and curve fitting.

E    Western blot and graph showing typical efficiency of Rab34 siRNA knockdown after 48 h in HeLa cells. Error bars show SEM from 3 experiments, ***P* < 0.001 (two-tailed *t*-test).

F    Representative confocal fluorescence images showing LAMP1 (magenta) distribution in HeLa cells under normal growth or starvation conditions in cells depleted of Rab34 or control cells transfected with a non-targeting siRNA. Giantin is shown in green. Scale bar, 10 μm.

G    Graphs showing cumulative distribution of LAMP1 intensity in normal growth or starvation conditions for control cells or cells depleted of Rab34 using siRNA. *P*-value is determined by the extra sum of F-squares test following nonlinear regression and curve fitting. Error bars show ± SEM from 30 cells in 3 replicates.

H    Graph showing percentage of LAMP1 in cis-medial Golgi region (giantin overlap) in the indicated conditions. Error bars show SEM from 30 cells in 3 replicates, ***P* < 0.001 (two-tailed *t*-test).

suppressed the ability of GFP-Rab34 to induce peri-nuclear lysosome accumulation (Fig 3C and D). To determine whether endogenous Rab34 itself contributes to starvation-induced peri-nuclear clustering of lysosomes, cells were depleted of Rab34 by siRNA (Fig 3E) and co-labelled with LAMP1 and Giantin (Fig 3F). Although the depletion of Rab34 did not strikingly affect lysosome distribution in normal growth conditions the peri-nuclear clustering of lysosomes was inhibited following starvation (Fig 3F and G).

We confirmed these observations using an alternative quantitative approach that directly measures lysosomes in the Golgi region by determining the proportion of LAMP1 fluorescence in the region of the cell defined by the cis/medial Golgi marker Giantin. Whilst this approach does exclude some of LAMP1 signal that visual observation would suggest is predominantly peri-nuclear, it does provide consistent frame of reference across the population. Both FLCN and Rab34 depletion suppressed a starvation-dependent increase in LAMP1/Giantin overlap and double knockdown of Rab34 and FLCN did not show any additive effect in either normal or starved conditions (Fig 3H). Together, these data show that Rab34 is regulated by nutrient status and controls lysosome distribution in a FLCN-dependent manner.

## Peri-nuclear Rab34-positive membranes contact lysosomes and reduce their motility

Immunofluorescence and electron microscopy observations suggest that Rab34 is primarily Golgi localised in HeLa [19,21]. Other reports, however, have suggested that at least in macrophages, Rab34 can also associate with phagosomes, and so perhaps endocytic structures including LE/lysosomes [57]. Following transfection, GFP-Rab34 was primarily localised in the peri-nuclear region in association with the Golgi, with GFP signal closely associated with cis and medial Golgi markers GM130 and giantin as well as TGN marker golgin-97 (Fig EV3) with some lower intensity fluorescence in a reticulate network extending away from here. Although lysosomes were clearly clustered in the region of the cell with high GFP-Rab34 fluorescence intensity, linescan analysis of fluorescence intensity did not show a correlative relationship between Rab34 and LAMP1 fluorescence signals suggesting that Rab34 is not recruited to lysosomes in these cells (Fig 4A). To determine whether this reflected the localisation of endogenous Rab34 and any effect of starvation, cells grown in normal media and starvation media were stained with a monoclonal Rab34 antibody and co-stained for LAMP1 (Figs 4B and C, and EV4A). Under nutrient replete conditions, Rab34 staining appeared relatively diffuse with some accumulation in the peri-nuclear region. Under starvation conditions where lysosomes were clustered, peri-nuclear Rab34 was enhanced and formed bright puncta and clusters strongly resembling the localisation observed for over-expressed GFP-Rab34 (compare bottom right panels of 4A and 4B). Closer examination of single confocal sections revealed a close association of endogenous Rab34- and LAMP1-positive puncta comparable to that observed for GFP-Rab34 (Fig 4C). Depletion of FLCN appeared to reduce both the number and intensity of Rab34 punta (Fig EV4B).

**Figure 4.   Peri-nuclear Rab34-positive membranes contact lysosomes.**

A    Representative confocal fluorescence image (left) showing methanol-fixed HeLa cell transfected with GFP-Rab34 and stained for LAMP1. Linescan analysis shows relative intensity of GFP and LAMP1 signals. Scale bar, 10 μm.

B    Maximum intensity projection images from a confocal Z-stack showing LAMP1 and endogenous Rab34 in P-M-fixed HeLa cells from normal and starved conditions. Whole cells and fields from which these panels are derived are shown in Fig EV4A. Boxes highlight regions shown in (C). Scale bar, 2 μm.

C    Single confocal sections from the Z-stack in (B). Orange arrows show close LAMP1/Rab34 association.

D    Analysis of Lysotracker-Red dynamics using PCC decay in HeLa cells transfected with either GFP or GFP-Rab34. Data are from 10 cells per condition. Error bars show ± SEM. *P*-value is determined by the extra sum of F-squares test following nonlinear regression and curve fitting.

E    Live-cell spinning disc confocal image series at 10-s intervals (from Movie EV3) showing GFP-Rab34-positive compartment making dynamic contacts with lysosomes. Scale bar, 1 μm.

F    Live-cell spinning disc confocal image series at 10-s intervals (from Movie EV4) of FLCN-GFP and mCherry-Rab34 in the peri-nuclear region of a FLCN-GFP/HA-FNIP2/mCherry-Rab34 transfected HeLa cell. Blue arrow highlights FLCN–Rab34 association moving on a linear trajectory. Yellow arrow highlights shorter saltatory movements. White arrow highlights dynamic associations between distinct FLCN-GFP and Rab34-positive structures. Scale bar, 1 μm.

G    Confocal image showing localisation of endogenous FLCN and Rab34 in starved HeLa cells. Scale bar, 10 μm.

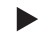

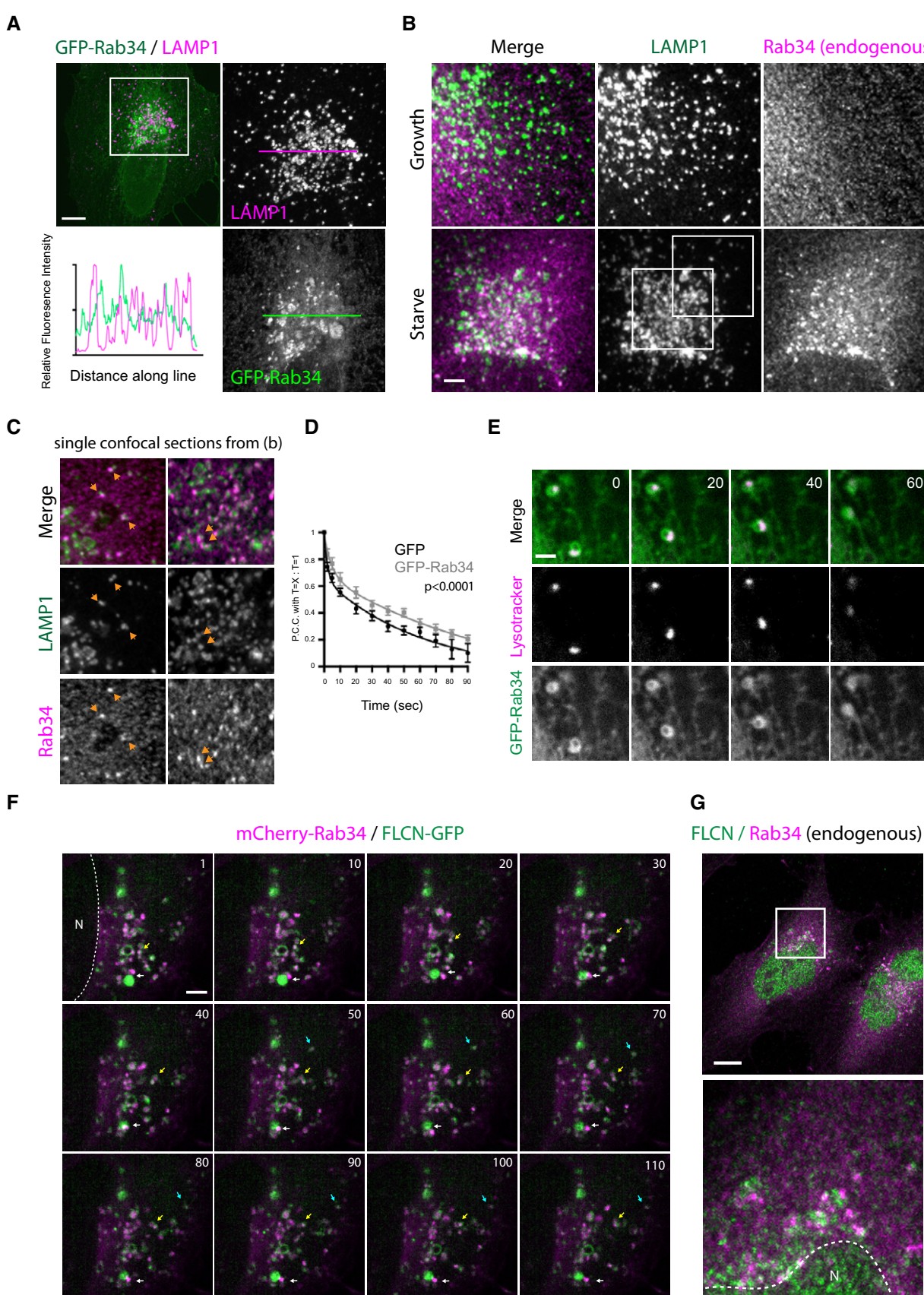

**Figure 4.**

Next, we analysed lysosome dynamics in Rab34 expressing cells by labelling acidic organelles with Lysotracker-Red. Consistent with our observations for LAMP1, Lysotracker-Red-positive puncta clustered around Rab34-positive membranes in the peri-nuclear region (Movie EV3). GFP-Rab34 expression also appeared to be associated with a reduction in the motility of the lysosomal compartment. To measure this, we compared the Lysotracker-Red fluorescence in the first frame of our movies with subsequent frames using the Pearson's correlation co-efficient (PCC; Fig 4D). The rate of PCC decay over time measures how quickly subsequent images deviate from the first frame and so gives an indication of the relative lysosome motility. This analysis showed a slower rate of Lysotracker-Red PCC decay in cells expressing GFP-Rab34 compared to GFP alone. At this resolution, the density of both Rab34 and Lysotracker-red in these regions of the cell made unequivocal identification of contact sites difficult. However, examination of GFP-Rab34 and Lysotracker-Red signals slightly more distal from this region appeared to show Rab34 accumulations, themselves associated with a larger reticulate network, interacting and moving with lysosomes (Fig 4E).

To examine the relationship between FLCN and Rab34, cells were transfected with mCherry-tagged Rab34 and FLCN-GFP/HA-FNIP2. Imaging of live cells expressing low levels of both proteins demonstrated dynamic association of FLCN-GFP with mCherry-Rab34 in the peri-nuclear region (Fig 4F and Movie EV4). We noted several different characteristics of mCherry-Rab34/FLCN associations including (i) colocalisation, overlap and comigration over short distances (blue arrow); (ii) dynamic association and saltatory movements (yellow arrow); (iii) dynamic, transient contacts between distinct accumulations of both proteins with little overlap (white arrow). To confirm that this low expression recapitulated the localisation of the endogenous proteins, HeLa cells cultured under starved conditions were stained for Rab34 and FLCN, where we observed similar associations of endogenous FLCN and Rab34 (Fig 4G).

Deconvolution and 3D volume rendering of images acquired using a spinning disc confocal microscope reinforced the notion that Rab34-positive membranes in the peri-nuclear region make an intimate network of contacts with lysosomes (Fig 5A). This was further validated by super-resolution structured illumination microscopy imaging that shows lysosomes decorating Rab34-positive membranes (Fig 5B). Finally, conventional thin-section electron microscopy revealed direct contact between lysosomes and peri-nuclear membranes (Fig 5C), and cryo-immuno-EM showed that GFP-Rab34 resides on membranes that make direct contact with lysosomes (Fig 5D).

### FLCN associates with mitochondrial targeted Rab34 in a manner dependent upon its DENN domain

We were unable to reliably detect a direct interaction between FLCN and Rab34 using purified proteins in pull-down experiments. Given the transient and dynamic nature of Rab34/FLCN contacts implied by our live-cell imaging, we considered that a cell based equilibrium technique might be more appropriate to address whether FLCN can interact with Rab34, if such an interaction is transient or of low affinity. We designed a construct to fuse Rab34, or a control, Rab35 (deleted of C-terminal lipidation sequences), C-terminally to a dsRED fluorophore followed by a peptide derived from *Listeria monocytogenes* ActA which directs the chimeric protein to mitochondria (Fig EV5A) [58]. HeLa were co-transfected with

FLCN-GFP/HA-FNIP2 and Rab34-dsRED-Mito or Rab35-dsRED-Mito. Robust recruitment of FLCN-GFP to mitochondria was observed in Rab34-dsRED-Mito cells but not in Rab35-dsRED-Mito cells (Figs 6A and EV5B). In contrast, HA-FNIP2 showed no obvious enrichment on mitochondria suggesting that this association is driven by FLCN itself. However, FLCN/FNIP2 dual-positive puncta (white arrows, presumably lysosomes) were still visible. Consistent with the notion that FLCN–Rab34 association does not require FNIP2, FLCN-GFP was also recruited to mitochondria in the absence of HA-FNIP2 co-expression (Fig EV5C, top). Following transfection, FLCN tends to be expressed a much higher level than FNIP proteins (Fig EV2D), and so given the nutrient-independent strong localisation of FNIP2 to lysosomes (Fig EV2C), it is likely that the mitochondrial bound FLCN represents the remaining non-FNIP bound cytoplasmic pool.

To determine role of the Rab34 nucleotide state, we engineered low nucleotide affinity T66N and GTP hydrolysis-deficient Q111L mutations in Rab34-dsRED-Mito [59]. Transfection of the Q111L mutant resulted in recruitment indistinguishable to that observed for wild-type Rab34 suggesting that FLCN may interact with Rab34 in the active state (Fig EV5C, bottom). However, the true nucleotide state of these ectopically targeted GTPases is ambiguous and we were unable to detect any expression of a T66N-Mito fusion protein despite deriving and sequencing multiple constructs. To determine the region of FLCN required for Rab34-dependent recruitment of FLCN to mitochondria, Rab34-dsRED-Mito constructs were co-transfected with plasmids encoding FLCN(N-343)-GFP (ΔDENN) or FLCN(344-C)-GFP (DENN only). We observed robust recruitment of FLCN(344-C)-GFP (DENN only) but not FLCN(N-343)-GFP (ΔDENN), showing that the FLCN C-terminal DENN domain is both necessary and sufficient to support Rab34–FLCN association (Fig 6B). We confirmed that FLCN associated with the exterior of mitochondria by examining cells using super-resolution structured illumination microscopy (SIM) imaging that highlighted the presence of ring-like FLCN-GFP around Rab34 mitochondria (Fig EV5D).

### The FLCN-DENN domain promotes association of Rab34 and RILP

Consistent with a capacity to associate with a complex including the active form of Rab34, in larger scale GST-RILP pull-down experiments from HeLa cells, endogenous FLCN was detected by Western blot, along with Rab34 and Rab7 (Fig 6C). To examine the regions of FLCN and RILP required for this association, the N-terminal coiled coil region of RILP (aa N-240), the Rab binding domain (aa 241–321) and the C-terminal domain of RILP (322 to C) were expressed in *E.coli*. Pull-down assays using extracts from UOK-FLCN (full length) and UOK-FLCNΔDENN cells (see Fig 7) showed some capacity for all three RILP fragments, but not GST alone to interact with full-length FLCN (Fig 6D). However, truncation of FLCN to remove the DENN domain eliminated any detectable association, further highlighting the importance of this region of FLCN for these interactions. Next, we examined whether the interaction between RILP and FLCN-DENN is direct. Consistent with this, GST-RILP, but not GST retained His-FLCN-DENN on beads from *E. coli* extract (Fig 6E). We tested whether the FLCN-DENN domain possesses GEF activity towards Rab34 *in vitro*, by examining the capacity of the purified protein to promote the GTP-dependent

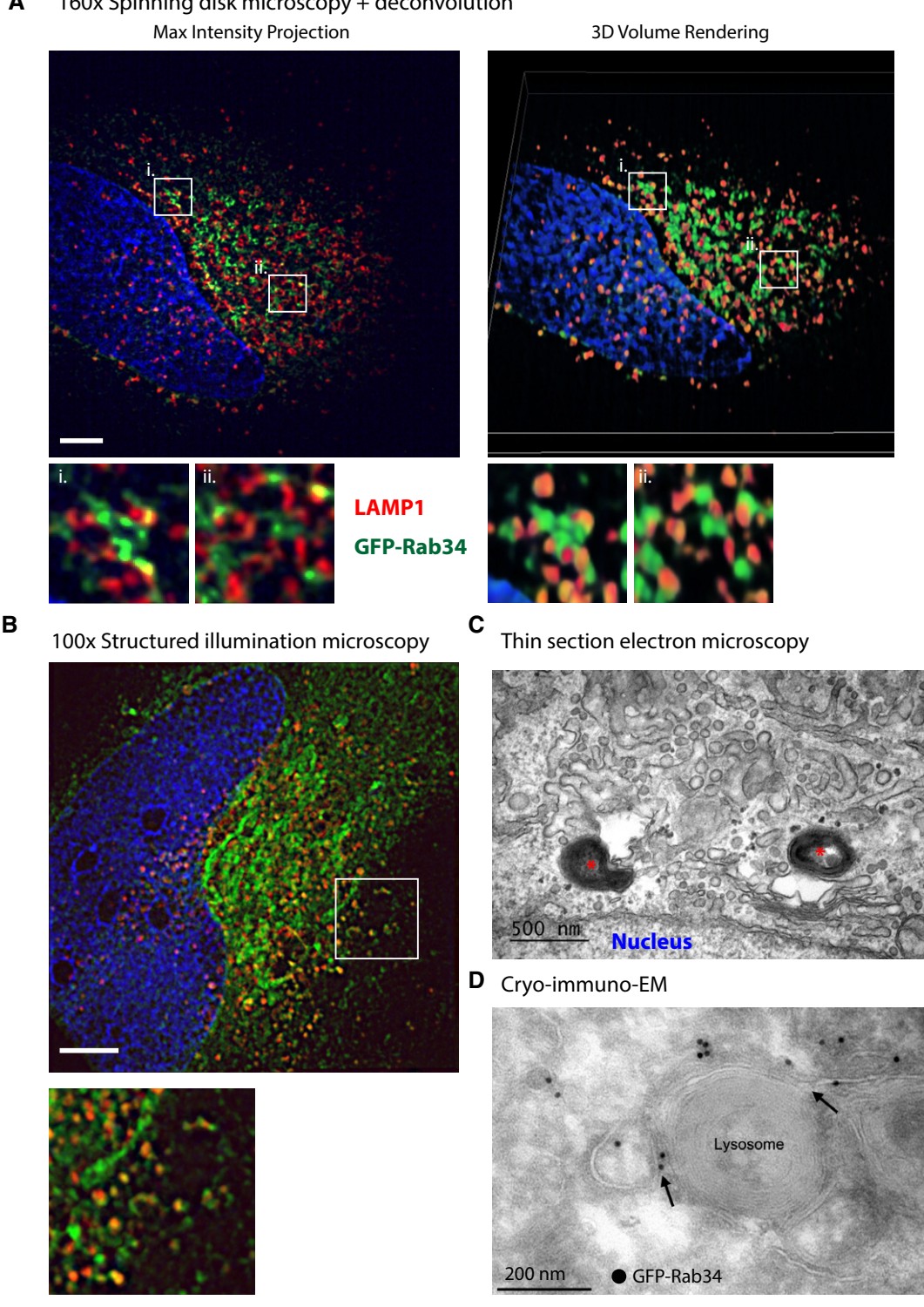

**A** 160x Spinning disk microscopy + deconvolution

Max Intensity Projection    3D Volume Rendering

LAMP1
GFP-Rab34

**B** 100x Structured illumination microscopy

**C** Thin section electron microscopy

500 nm    **Nucleus**

**D** Cryo-immuno-EM

Lysosome

200 nm    ● GFP-Rab34

**Figure 5.  High-resolution image analysis of Rab34 lysosome contacts.**
Images from fixed GFP-Rab34 transfected cells showing contacts between GFP-Rab34-positive peri-nuclear membranes and lysosomes.

A   Spinning disc confocal microscopy images acquired at 160× magnification. Left panels show a maximum intensity projection imaged of a Z-stack, sampled according to the Nyquist criterion and deconvolved. Right panels show a 3D volume rendering of the same data. Scale bar, 2 μm.
B   A projection of 4 planes from a structured illumination microscopy super-resolution Z-stack of the same cells. Scale bar, 2 μm.
C   A thin-section electron microscopy image of the peri-nuclear region of a GFP-Rab34 transfected HeLa cell. Asterisks (*) indicate lysosomes.
D   A cryo-immuno-EM image of a GFP-Rab34 transfected HeLa cell. GFP-Rab34 (dark black dots) is identified by immunostaining for GFP and detection with gold conjugated protein A. The lysosome is intimately associated with Rab34-positive membranes. Sites of contact are highlighted by arrows.

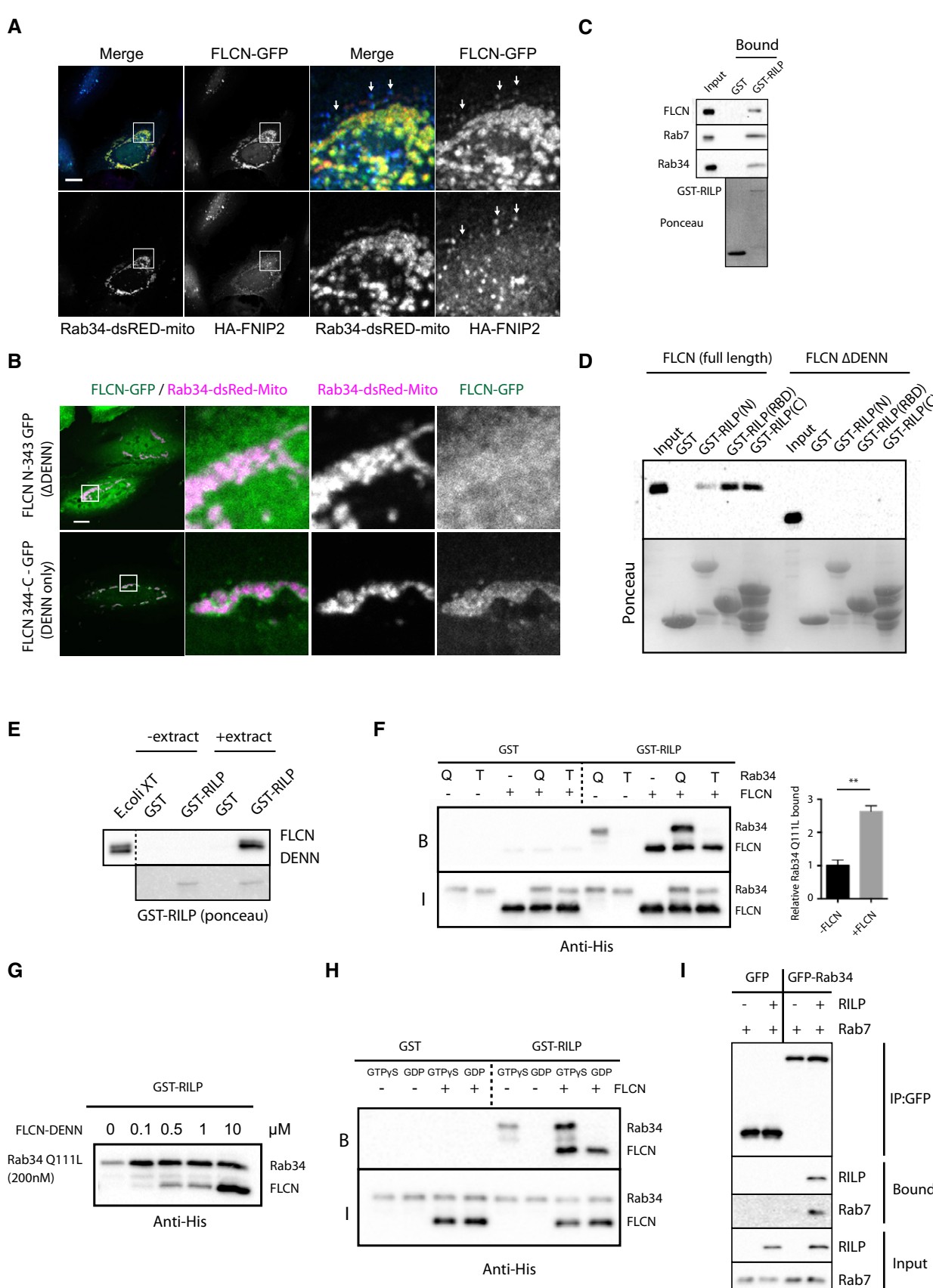

**Figure 6.**

**Figure 6.  FLCN directly promotes the association of Rab34 and RILP.**

A   Confocal immunofluorescence images of HeLa cells transfected with Rab34-dsRED-Mito, FLCN-GFP and HA-FNIP2. White arrows highlight FLCN-GFP/HA-FNIP2 co-localisation distinct from Rab34-mitochondria/FLCN-GFP localisation. Scale bar, 10 μm.
B   Confocal immunofluorescence images of Rab34-dsRED-Mito (WT) and FLCN-GFP DENN only or ΔDENN transfected HeLa cells showing DENN domain-dependent recruitment to mitochondria. Scale bar, 10 μm.
C   Western blot analysis of GST-RILP pull down showing association of endogenous FLCN, Rab34 and Rab7 with RILP. Input lane shows 1% input cell extract.
D   Western blot analysis of GST-RILP pull down from UOK257 cells showing DENN domain-dependent association of FLCN with RILP N-terminal, Rab binding domain (RBD) and C-terminal fragments.
E   Western blot showing binding of His-tagged FLCN-DENN domain in *E. coli* extract to GST-RILP.
F   Western blot analysis of GST-RILP pull down showing direct interaction between RILP and GTP hydrolysis-deficient (Q111L) Rab34 (at 200 nM, in the presence of 1 mM GTPγS) and FLCN-DENN domain (1 μM), but not low nucleotide affinity (T66N) Rab34. FLCN-DENN enhances association of RILP with Q111L Rab34. Graph shows quantification of relative RILP/Rab34 Q111L binding. Error bars show ± SEM. **$P < 0.01$ (two-tailed *t*-test).
G   Western blot analysis of GST-RILP/Rab34 Q111L pull down (Rab34 at 200 nM) showing that enhanced association of Rab34 with RILP is achieved with low concentrations of FLCN.
H   Western blot analysis of GST-RILP pull-down experiments with GTPγS loaded (in the presence of 1 mM GTPγS) and GDP loaded (in the presence of 1 mM GDP) wild-type Rab34, showing FLCN-DENN (500 nM)-induced interaction with RILP.
I   Western blot analysis of a GFP-Trap immunoprecipitation experiment from HeLa cells showing that expression of HA-RILP promotes the association of GFP-Rab34 and HA-Rab7.

dissociation of the fluorescent GDP analogue mant-GDP from purified Rab34 [60] (Fig EV6A and B). Proper mant-GDP loading and sensitivity of the assay was confirmed by monitoring decay in fluorescence induced by the $Mg^{2+}$ chelating agent, EDTA, that promotes nucleotide dissociation (Fig EV6A, orange line) [60]. However, consistent with [28], even at high concentrations of FLCN-DENN (10 μM) we did not detect any GEF activity (Fig EV6A, left graph). Similarly, nucleotide exchange was not promoted by addition of RILP (Fig EV6A, middle graph). As FLCN and RILP interact, we considered the possibility that FLCN may act as a GEF in the presence of RILP; however, titration of FLCN up to 10 μM in the presence of 1 μM RILP also failed to induce GEF activity (Fig EV6A, right graph). We considered the alternative hypothesis that FLCN may control the interaction of active, GTP bound, Rab34 and RILP. To test this directly, we performed GST-RILP pull-down experiments using purified recombinant proteins (200 nmol RILP on beads, Rab proteins at 200 nM and FLCN-DENN at 1 μM) (Fig 6F). As expected, RILP bound both purified FLCN-DENN (lane 8) and the hydrolysis-deficient Rab34 Q111L mutant (in the presence of the non-hydrolysable GTP analogue GTPγS) (lane 6), but not the low nucleotide affinity T66N mutant (in the presence of GDP) (lane 7). In the presence of FLCN-DENN, however, RILP binding to Rab34 Q111L was enhanced by approximately 2.7-fold (lane 9), without promoting binding of

the T66N mutant (lane 10). We examined the concentration dependence of this effect. Maintaining the concentration of Rab34 Q111L at 200 nM but titrating FLCN-DENN from 0 to 10 μM revealed that enhanced Rab34 binding is achieved with a low concentration of FLCN (100 nM) and is not further increased by concentrations of FLCN that are up to 2 orders of magnitude higher (Fig 6G). We also noted that at low concentrations of FLCN-DENN, considerably more Rab34 was retained than FLCN. This suggests that FLCN facilitates the loading of active Rab34 onto RILP, rather than requiring the formation a stable tri-partite complex. We confirmed that these observations were not restricted to Q111L and T66N mutant forms of Rab34 by loading 200 nM wild-type Rab34 with GTPγS or GDP, which again showed that the FLCN-DENN (0.5 μM) domain directly promotes the formation of an active Rab34–RILP complex (Fig 6H).

Rab7 and Rab34 likely to bind RILP using similar mechanisms [19], however, as RILP can form a dimer [61,62] our data suggested the possibility that it may serve to link Rab34 on Golgi with Rab7 on lysosomes. Therefore, we tested whether RILP can promote the association of Rab34 and Rab7. GFP-Rab34 and HA-tagged Rab7 were expressed in cells with or without co-transfection of HA-tagged RILP and binding to Rab34 was assessed by GFP-TRAP immunoprecipitation experiments and Western blot analysis. In the absence

**Figure 7.  Folliculin controls lysosome distribution and dynamics in BHD kidney cancer cells.**

A   Western blot showing endogenous Rab7 and Rab34 expression in UOK257 and UOK257-2 cell extracts and relative levels of the active forms as measured by GST-RILP pull down. HSC70 is used as a loading control. Graphs show quantification of bound Rab7/Rab34 from 3 independent experiments, error bars show SEM, *$P < 0.05$ (two-tailed *t*-test).
B   Representative confocal immunofluorescence images (top) showing intracellular distribution of LAMP1 in UOK257 and UOK257-2 cells. Dashed line highlights cell periphery in UOK257-2 cells. Scale bar, 10 μm. Graphs showing cumulative distribution of LAMP1 intensity in UOK257 vs UOK257-2 cells (bottom left) and Lysotracker-Red PCC decay with time from confocal movies of UOK257 and UOK257-2 cells (bottom right). Error bars show ± SEM from 30 cells in 3 replicates (left) or from 5 cells (right). *P*-value is determined by the extra sum of F-squares test following nonlinear regression and curve fitting.
C   Widefield immunofluorescence images showing early endosomes (EEA1) and Golgi (golgin-97) in UOK257 and UOK257-2 cells. Scale bar, 10 μm.
D   Western blot showing FLCN and FLCNΔDENN expression in stable UOK257 derived cell lines.
E   Representative widefield fluorescence images showing intracellular distribution of LAMP1 in UOK257, UOK257-FLCN and UOK257-FLCNΔDENN cells. Scale bar, 10 μm. Graph showing cumulative distribution of LAMP1 intensity in UOK257, UOK257-FLCN and UOK257-FLCNΔDENN cells. Error bars show ± SEM from 30 cells in 3 replicates. *P*-value is determined by the extra sum of F-squares test following nonlinear regression and curve fitting.
F   Western blot (left) showing endogenous Rab7 and Rab34 expression in UOK257, UOK257-FLCN and UOK257-FLCNΔDENN cells and relative levels of activity as measured by GST-RILP pull down. HSC70 is used as a loading control. Graphs show quantification of bound Rab7/Rab34 (right) from 3 independent experiments (*$P < 0.05$, ***$P < 0.001$, two-tailed *t*-test).
G   Western blot showing endogenous RILP expression in the indicated cell lines. Graph shows quantification of RILP expression from 3 independent experiments (*$P < 0.05$, **$P < 0.01$, two-tailed *t*-test).

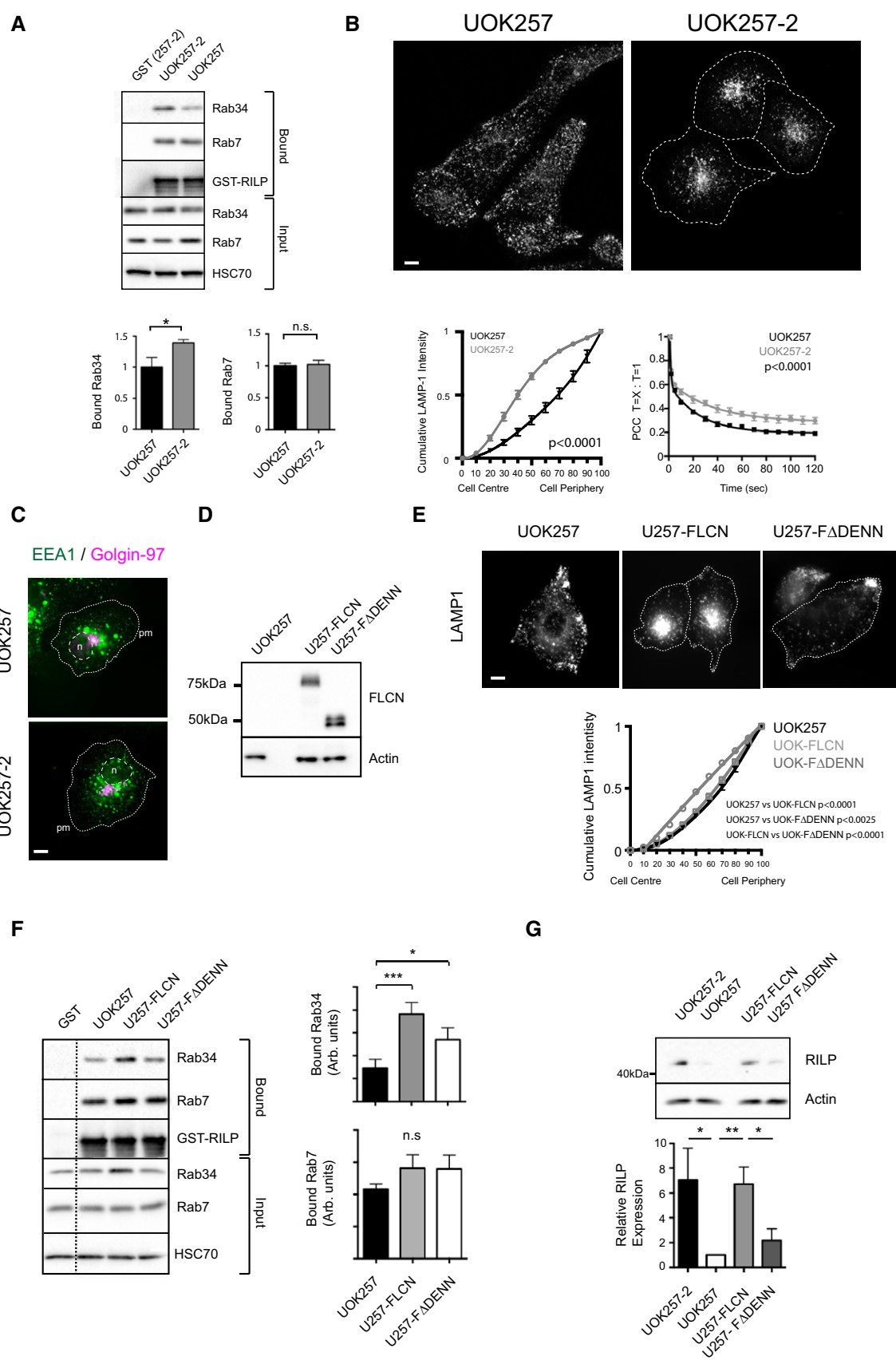

**Figure 7.**

of RILP, Rab7 did not form a complex with Rab34, however, in the presence of RILP, Rab7 co-immunoprecipitated with the Rab34–RILP complex, demonstrating that RILP can promote the association of the two small GTPases (Fig 6I).

### UOK257 (FLCN deficient) and UOK257-2 (FLCN restored) kidney cancer cells exhibit distinct lysosome distributions and dynamics

Next, we examined Rab34 and lysosome distribution in the FLCN-null BHD kidney cancer cell line UOK257 and its clonal derivative UOK257-2 where FLCN expression is restored [45,63]. Western blot analysis of cell extracts and GST-RILP pull-down analysis revealed a small but reproducible increase in Rab34 expression that was also associated with a significantly higher Rab34 retention by GST-RILP from extracts prepared from UOK257-2 cells compared to UOK257 cells (Fig 7A). In contrast, on the same resin, we detected no significant difference in the amount of Rab7 pulled down by RILP. Staining for LAMP1 revealed that in UOK257-2 cells, lysosomes were predominantly concentrated within a peri-nuclear cluster. In contrast, lysosomes in FLCN-null UOK257 cells were dispersed throughout the cytoplasm (Fig 7B, top and bottom left panels). The distribution of the early endosome marker EEA1 and the TGN marker Golgin-97 were comparable in both cell lines (Fig 7C). We generated UOK257 and UOK257-2 cells by lentiviral transduction and FACS sorting that stably expressed the LE/lysosome marker GFP-Rab7 at low levels and identified acidic organelles using Lyso-tracker-Red (Movies EV5 and EV6). Live confocal imaging confirmed the differential lysosome distribution observed in fixed cells for LAMP1 and also revealed striking differences in their dynamic behaviour. In UOK257 cells lacking FLCN, lysosomes were highly dynamic, moved rapidly throughout the cytoplasm on linear trajectories, and transited in and out of prominent accumulations at the cell periphery (Movie EV5). In contrast, the motility of lysosomes in UOK257-2 cells appeared spatially constrained with the majority making only short saltatory movements in the peri-nuclear region (Movie EV6). The smaller subset of lysosomes outside of this region, however, did make longer linear runs. Maximum intensity time projection images of Movies EV5 and EV6 illustrate these differences in dynamics and distribution by highlighting the area occupied by Rab7 and Lysotracker-Red fluorescence over the duration of the entire movie (Fig EV7). PCC decay analysis for Lysotracker-Red confirmed this reduction in lysosome dynamics in FLCN expressing cells (Fig 7B, bottom right panel).

Re-expression of full-length FLCN in UOK257 (UOK257-FLCN) (Fig 7D and E) restored a predominantly peri-nuclear lysosome distribution, although quantitatively, the extent of redistribution to the peri-nuclear region was slightly less than that observed in UOK257-2 cells (Fig 7E). Rab34 expression and the amount of Rab34 retained by GST-RILP was also increased (Fig 7F). In contrast, a FLCN variant lacking the C-terminal DENN domain expressed at a comparable level failed to promote peri-nuclear accumulation (UOK257-FΔDENN) (Fig 7D–F). We also noted that expression of endogenous RILP was also low in UOK257 cells compared to UOK257-2, but was restored in a DENN domain-dependent manner in FLCN over-expressing cells (Fig 7G). Taken together, these data are strongly supportive of a DENN domain-dependent role for FLCN regulation of lysosome distribution through a Rab34/RILP axis in BHD kidney cancer cells.

## Discussion

Since the identification of FLCN as a tumour suppressor whose disruption is responsible for BHD syndrome, considerable effort has been focused on understanding the underlying molecular mechanisms that lead to disease. Recent studies demonstrating that the lysosomal surface is a key site of action for FLCN have provided particularly crucial insight, showing that the FLCN/FNIP complex can interact with Rag GTPases, supports the amino acid-dependent recruitment of mTORC1 to the lysosomal surface and can regulate the nuclear/cytoplasmic localisation of lysosome associated transcription factors [37–39,52]. Our data build on these studies and demonstrate a new, unanticipated, function for FLCN at the lysosome as a direct connection between the lysosomal nutrient signalling network and the cellular machinery that controls the intracellular distribution of the organelle itself.

We show that FLCN and Rab34 promote the peri-nuclear lysosome clustering following nutrient withdrawal. Control of lysosome distribution requires the FLCN-DENN domain, which we demonstrate associates with Rab34 in cells and interacts directly with the Rab34 effector RILP. *In vitro*, rather than acting as a GEF for Rab34, the FLCN-DENN domain promotes the loading of active Rab34 onto RILP. A series of high-resolution image analyses demonstrate the Rab34-positive peri-nuclear membranes make direct contacts with lysosomes in this region of the cell that result in a restriction of lysosome motility.

Our data provide a new mechanistic and regulatory context for the established observation that Rab34 promotes the RILP-dependent peri-nuclear clustering of lysosomes whilst residing on a non-lysosomal compartment [19,21]. Moreover, our studies in BHD kidney cancer cells suggest that this pathway may play a role in the pathogenesis of BHD syndrome. A recent report examining antigen presentation in dendritic cells also suggest Rab34 may respond to other extrinsic stimuli (in this case LPS) to promote context-dependent changes in lysosome distribution that modulate lysosome fusion with phagosomes [64].

We propose a model where starvation-induced FLCN association with lysosomes drives dynamic, Rab34/RILP driven interactions between lysosomes and non-lysosomal peri-nuclear membranes that result in limitation of their motility, promote peri-nuclear retention and thus contribute to control of their cytoplasmic distribution. We do not exclude a role for Rab7 itself in this process despite the limited over-expression phenotype. Given that Rab7 is thought to be responsible for the recruitment of RILP to lysosomes, we speculate, as others have [21], that these contacts may result from a formation of a Rab34–RILP–Rab7 platform mediated by dimeric RILP [61,62]. The notion that RILP functions at inter-organelle contacts is support by [18], who show that the cholesterol sensor ORP1L acts through the ER protein VAP, in trans, to control RILP interaction with the dynein–dynactin complex.

Here, we have focused our studies primarily on FLCN itself, but it is clear that FLCN requires FNIP proteins to associate with lysosomes [37] and FNIP depletion also inhibits starvation-induced clustering. However, FNIP driven FLCN lysosome association is not sufficient to induce peri-nuclear clustering in the absence of starvation, but can induce the formation of dynamic tubules that extend from lysosomes. Thus, it may be that a second starvation-dependent signal is also required to direct clustering. This may require the

modulation of lysosome association or activity of Arl8 to initiate a minus end directed bias in microtubule transport [2]. Further studies should investigate whether FLCN/FNIP regulates the BORC/Arl8/SKIP/kinesin-1 axis that promotes lysosome dispersion [4,8].

It is interesting to note that the *S. cerevisiae* homologue of FLCN, Lst7, lacks the C-terminal DENN domain that is crucial for FLCN function in this context. Yet Lst7 interacts with the FNIP homologue Lst4, is capable of nutrient-dependent association with the vacuole and carries out an analogous role in the mTORC1 pathway [29,42]. Neither do yeast have a RILP homologue or close homologue of Rab34. Given the very different organisation of late endocytic/vacuolar compartments in yeast, it may be that the FLCN-DENN domain is associated with organisms that have a greater requirement to spatially regulate distribution of these compartments over long distances on microtubules.

The functional relevance of our mechanistic studies in HeLa is supported by our experiments in BHD kidney cancer UOK257/UOK257-2 cells, although there is some divergence, not least in the apparent nutrient independence of differential lysosome distribution in these BHD cancer lines. This may be due to the complete long-term loss of full-length FLCN in these cells when compared to acute depletion, their tissue origin or metabolic adaptions acquired during tumourigenesis and clonal selection [47]. It is notable that UOK257 cells do not show large deficiencies in mTORC1 activity [33]. It will therefore be useful to expand these studies to other cell types including renal and lung epithelia to understand how FLCN/Rab34-dependent changes in lysosome dynamics may contribute to BHD syndrome. Nonetheless, given the complex relationship between lysosome dynamics, autophagy and mTORC1 activity [2], and emerging intimate and equally complex connections between FLCN and the same pathways [37,38,49], we suggest that mis-regulation of lysosome dynamics/distribution by disruption of FLCN may contribute to the highly context-dependent autophagy and mTORC1 activity phenotypes found in various BHD model systems studied.

In summary, we have identified a new, direct, mechanistic connection between the lysosomal nutrient signalling network and the cellular machinery that controls the intracellular distribution of the organelle itself. Further studies should focus on understanding the regulatory mechanisms that control this process.

## Materials and Methods

### Antibodies, plasmids and reagents

Plasmids encoding human FLCN-HA and HA-FNIP-2 were kind gift from L.Schmidt (NIH). To express full-length hsFLCN or hsFLCNΔDENN (aa1-343) in an untagged form by lentiviral vector, the coding sequence of FLCN was amplified by PCR and subcloned into pLVX (Clontech). To express FLCN or variants (ΔDENN) and DENN only) with a C-terminal GFP-tag full-length mouse FLCN, or sequences encoding amino acids N-343 or 344-C were amplified from I.M.A.G.E clone 5099493 and subcloned into a CMV driven expression vector (pCB6-cGFP). GFP-Rab7 in a mammalian expression vector was obtained from the Addgene collection (12605) and for GFP-tagged lentiviral expression, coding sequences of Rab7 were amplified by PCR and subcloned into vector pLL-nGFP. For expression HA-Rab7, the Rab7 ORF was subcloned into vector CB6-HA. A

mammalian expression plasmid encoding human GFP-RILP was a gift from C. Bucci (University of Salento). myc-SKIP was a gift from S. Munro (MRC-LMB, Cambridge). For bacterial expression of RILP, the full ORF or the indicated fragments were subcloned into a bacterial expression vector with an N-terminal GST tag, or for HA-tagged expression in mammalian cells, the same sequence was subcloned into pCB6-HA. Mouse Rab34 and Rab35 were obtained as an I.M.A.G.E. clones. Rab34 was subcloned into a mammalian expression vector with an N-terminal GFP tag (CB6-nGFP), an N-terminal mCherry tag (CB6-Cherry) or the bacterial expression vector pET28 His-thrombin. Human FLCN-DENN domain for bacterial expression (amino acids 341–556) was subcloned into the same vector. For mitochondrial targeting, Rab34 and Rab35 were subcloned into pCAG-ds-RED-Mito based upon [58] without their C-terminal cysteine residues (ΔCCP – Rab34, ΔCC – Rab35). Mutations were introduced by site-directed mutagenesis and all constructs were verified by sequencing. FLCN siRNA was supplied by Thermoscientific (On target PLUS Pool L-009998-02, sequences 05 and 08 from this pool were used as single oligos and named oligo 1 and 2, respectively). FNIP1/2 siRNA was obtained from the same supplier (L-032573-00 and L184611-00). Rab34 siRNA was J-009735-05 and J-009735-08, named oligo 1 and 2, respectively. Rabbit anti-FLCN (D14G9), anti-Rab7 (D95F2), anti-pS6K (108D2), anti-p4EBP, anti-EEA1 (C45B10) and anti-LAMP1 (D2D11) were obtained from Cell Signaling Technologies. Rabbit anti-RILP (sc-98331, for WB), mouse anti-HSC70 (clone B6), mouse anti-Golgin 97 (CDF4) and mouse anti-Rab34 (sc-376710, clone C-5) (used for IF) were obtained from Santa Cruz Biotechnology. Mouse anti-GM130 (35/GM130) was from BD Biosciences. Anti-GFP (3E1) and anti-actin (AC-74) were obtained from Roche and Sigma, respectively. Rabbit Anti-Rab34 (ab73383) (used for Western blot) and mouse anti-mitochondria (MTC02) were supplied by Abcam. Mouse anti-LAMP1 (H4A3) was obtained from the Developmental Studies Hybridoma Bank.

### Cell culture, transfection and nutrient withdrawal

All cells were grown in DMEM supplemented with 10% FCS, Glutamine and Penicillin/Streptomycin (Normal growth in figures and text). HeLa cells were transfected using Effectene or Hiperfect for DNA plasmids or siRNA, respectively, when indicated (Qiagen). Serum and amino acid starvation was carried out by washing cells twice in Krebs-Ringer Bicarbonate buffer solution plus 1 mg/ml Glucose, pH 7.4 (Sigma) and incubating in buffer in all cases for 4 h prior to the indicated analysis. Acidification treatments were carried out as described by Heuser [54].

### Bacterial protein expression and purification

For expression and purification of GST-RILP or GST-RILP fragments described above, plasmids were used to transform *E. coli* BL21 (DE3). Single colonies were picked and grown at 37°C overnight. Small-scale overnight bacterial cultures were used to inoculate 1 l cultures that were incubated at 37°C until they reached an OD600 of 0.5. The temperature was then lowered to 16°C and protein synthesis was induced by the addition of 300 μM IPTG for 16 h. Cells were harvested by centrifugation at 5,000 *g* for 15 min at 4°C and resuspended in 25 mM HEPES pH 7.5, 500 mM NaCl and 5 mM β-mercaptoethanol supplemented with protease inhibitor cocktail

(Roche). Cell lysis was accomplished by sonication. Insoluble material was sedimented by centrifugation at 16,500 $g$ for 1 h at 4°C and the supernatant filtered using a 0.22 μm filter prior to incubation with 1 ml Glutathione coated beads. After extensive washing, samples were eluted with 10 mM Glutathione and dialysed into the glutathione free buffer prior to snap freezing and storage at −80°C. His-tagged proteins were expressed in a similar manner, but in the presence of 25 mM imidazole and in the case of Rab proteins, 2 mM MgCl$_2$, isolated using His-trap FF columns (GE Life Sciences) and eluted with an imidazole gradient using an AKTA Prime system (GE Life Sciences). Proteins were dialysed overnight against imidazole-free buffer, snap frozen and stored at −80°C. Protein concentration in solution was determined using a nanodrop spectrophotometer.

### Rab34 GTPase loading and GEF assays

Rab34 proteins were dialysed overnight into loading buffer consisting of 25 mM HEPES pH 7.5, 150 mM NaCl and 5 mM EDTA. Nucleotide was loaded (mant-GDP (Thermoscientific), GDP or GTPυS) by addition of a 25-fold molar excess of the nucleotide for 2 h at room temperature followed by addition of 10 mM MgCl$_2$. Non-bound mant-GDP was removed using a PD-10 buffer exchange column (GE Life Sciences) equilibrated in exchange buffer HEPES pH 7.5, 150 mM NaCl, 2 mM MgCl$_2$. For GEF assays, 300 nM Rab34 (in a volume of 100 μl) loaded with mant-GDP was incubated with various combinations and concentrations of His-FLCN-DENN and GST-RILP. Reactions were performed in black non-binding coated 96-well plates (Greiner), and mant-GDP fluorescence was monitored using 355 nm excitation and 460 nm emission ± 10 nM band pass filters on a BMG Polarstar Omega plate reader. GTP was added at a concentration of 0.3 mM or EDTA at a concentration of 10 mM at the 2-min time point. Data were acquired at 10-s intervals for 25 min. Curves are mean of duplicate samples and are smoothed using a 6-point rolling average and polynomial fitting to reduce small fluctuations in signal due to instrument noise.

### GST-RILP pull-down assays

For pull down of cellular proteins, $1.5 \times 10^6$ cells were seeded into 10 cm dishes the day prior to the experiment. Cells were treated as indicated and then washed with ice-cold wash buffer (25 mM HEPES pH 7.5, 150 mM NaCl, 2 mM MgCl$_2$) and lysed with 750 μl lysis buffer (wash buffer plus 0.2% NP-40, 0.2% Triton X-100 containing a protease inhibitor cocktail, Roche) for 10 min prior to centrifugation at 13,000 $g$ for 10 min at 4°C; 50 μl of supernatant was retained for analysis of input levels (20 μl loaded), and the remainder was incubated with 20 μl glutathione coated beads with 0.1 nmol prebound GST-RILP. Beads were washed 4× and boiled in 40 μl SDS-loading buffer (20 μl loaded). Samples were analysed by Western blot using the indicated antibodies. For larger scale pull downs required to detect association of endogenous FLCN with RILP, 0.5 nmol GST-RILP was used along with $2 \times 15$ cm dishes of HeLa lysed in 1 ml lysis buffer. To detect interactions of purified recombinant proteins, 0.2 nmol GST-RILP proteins were bound to glutathione coated beads and incubated with the indicated proteins in a volume of 300 μl in buffer comprising HEPES pH 7.5, 250 mM NaCl, 2 mM MgCl$_2$ and where indicated 1 mM GDP or GTPγS for 14 h at 4°C. 5 μl input proteins were retained for analysis. After

incubation, beads were washed 4× in binding buffer, and resuspended in 40 μl SDS-loading buffer (10 μl loaded). Bound and input proteins were analysed by Western blot using an anti-His tag antibody. Quantifications presented are mean of 3 independent experiments with two-tailed $t$-test used for statistical analysis.

### GFP-TRAP immunoprecipitation

$1 \times 10^6$ HeLa cells were plated on 10 cm dishes and transfected with the indicated combinations of GFP, GFP-Rab34, HA-Rab7 and HA-RILP. After 16 h, transfected cells were lysed in 1 ml of 25 mM HEPES pH 7.5, 150 mM NaCl, 2 mM MgCl$_2$, 0.5% NP-40, 0.5% Triton X-100 containing a protease inhibitor cocktail (Roche) for 10 min prior to centrifugation at 13,000 $g$ for 10 min at 4°C. The resulting supernatant was incubated with 20 μl of prewashed GFP-Trap (Chromotek) beads for 90 min. Beads were washed 4× in assay buffer (25 mM HEPES pH 7.5, 150 mM NaCl, 2 mM MgCl$_2$), resuspended in 40 μl of buffer and 15 μl of SDS-loading buffer before boiling; 10 μl of samples was subjected to SDS–PAGE and analysed by Western blot using antibodies against GFP and HA; 10 μl of total cell lysate was loaded for analysis of input levels.

### Western blot quantification

Non-saturated chemi-luminescent images were acquired using a Bio-Rad XR system, and band intensities were analysed with Image-lab software.

### Immunofluorescence

$1 \times 10^5$ cells were plated onto fibronectin coated coverslips in 6-well plates and transfected with various plasmids or siRNA. Plasmids were transfected 16 h prior to analysis, siRNA 48 h prior to analysis. Cells were starved when indicated and fixed with either PFA or −20°C methanol before blocking and probing with primary and secondary antibodies (Alexa 488, 568 and 633 conjugated anti-mouse or anti-rabbit secondary antibodies, Thermoscientific). Widefield fluorescence images were collected using a Zeiss Olympus IX-81 microscope with either a 40× or 100× objective running Metamorph. Confocal images were collected using a Nikon A1 system with a 100× objective running NIS Elements. Super-resolution imaging was performed an N-SIM Super resolution system. The spinning disc system is described below.

### Live-cell imaging and analysis

For live-cell imaging, $1 \times 10^5$ the indicated cells were plated and transfected in fibronectin coated 35 mm Mattek dishes. The following day, if required, lysosomes were labelled with 20 nM lyso-tracker-Red and cells were imaged at a rate of 1 frame per second either using an inverted Nikon A1 confocal system with a 100× objective lens or an inverted CSU-X1 Spinning Disk Confocal system with an Andor Ixon3 EM-CCD camera and a 100× objective lens and when required a 1.6× auxiliary magnifier, both equipped with temperature and $CO_2$ control and running NIS Elements. Movies were processed using NIS elements and Image J. Figures were assembled using Image J in conjunction with Adobe Photoshop and Illustrator packages (Adobe, CA, USA). Correlation analysis was

performed by extracting individual frames, thresholding to remove background and applying the Image J Intensity Correlation Analysis plugin, comparing the first frame of the movie with sequential frames at the indicated time increments [65].

### Lysosome distribution analysis

To quantify lysosome distribution in an unbiased fashion, widefield images of cells were acquired at 40× magnification. The cell perimeter was defined by thresholding equivalent saturated images and the cell area was scaled in 10% decrements using ImageJ. After background subtraction, cumulative integrated LAMP1 intensity (relative to the whole cell) was then plotted for increasing incremental deciles. Data points are from a minimum of 30 cells in 3 replicates and are representative of at least 3 independent experiments. For analysis of the resulting data, the nonlinear regression function in Graphpad Prism was used to fit a centred 6[th] order polynomial. To compare models so assess the statistical significance of differences in distribution profiles, the extra sum of F-squares test was applied. *P*-values for particular comparisons are indicated on the graphs. As an alternative quantitative approach, the boundary of the region of the cell defined by the Cis/Medial Golgi marker Giantin was defined using ImageJ and the percentage of total cellular LAMP1 fluorescence intensity falling within this region was determine. Data represent mean percentages from 30 cells in 3 replicates, and two-tailed *t*-test was applied for statistical analysis. In cells where GFP-Rab34 was over-expressed, to confirm that populations with comparable expression levels were selected for quantification, cells were analysed that expressed Rab34 robustly but not at an excessively high level. In our system, these cells had average raw fluorescence intensities (over the whole cell) from the same exposure time in the range of 3,000–10,000 grey values (in 16-bit images). We confirmed that we had selected a comparable population for each oligonucleotide by comparing the mean of these values.

### Electron microscopy

For conventional EM, cells transfected with GFP-Rab34 on thermanox coverslips (NUNC) were fixed in 2% paraformaldehyde (PFA)/2% glutaraldehyde for 30 min, post-fixed in 1% osmium tetroxide, 1.5% potassium ferricyanide, incubated in 1% uranyl acetate, dehydrated and embedded in TAAB-812 resin. For cryo-immuno-EM, cells were fixed with 4% paraformaldehyde and 0.1% gluteraldehyde in 0.1 M phosphate buffer at pH7.4, infused with 2.3 M sucrose and supported in 12% gelatin. Sections (70 nm) were cut at −120°C and collected in 1:1 2.3 M sucrose:2% methylcellulose. Sections were incubated with rabbit anti-gfp antibody followed by 10 nm protein A gold (UMC Utrecht) and viewed on a Jeol 1010 TEM. Images were gathered using a Gatan OriusSC100B charge-coupled device camera.

**Expanded View** for this article is available online.

### Acknowledgements

We thank Professors Maddy Parsons and Anne Ridley for critical reading of the manuscript as well as Drs Chris Molenaar and Daniel Matthews (Nikon Imaging Centre, KCL) for assistance with image acquisition and analysis. This work was funded by a Wellcome Trust Research Career Development Fellowship to M.P.D. (WT097316/Z/11/Z). Y.Y.Y. is supported by a British Biotechnology and Biological Sciences Research Council project grant to M.P.D. (BB/L006774/1). E.R.E. is support by the Medical Research Council (166002).

## Author contributions

GPS, YYY, AS, PEM, ERE and MPD performed experiments and analysed data. ERE and MPD designed experiments. MPD wrote the paper.

## Conflict of interest

The authors declare that they have no conflict of interest.

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
