## [Review Process File · EMBO Reports]

Manuscript EMBO-2015-41382

Folliculin directs the formation of a Rab34-RILP complex to control the nutrient-dependent dynamic distribution of lysosomes

Georgina Starling, Yan Yip, Anneri Sanger, Penny Morton, Emily Eden and Mark Dodding

Corresponding author: Mark Dodding, King's College London

Review timeline:	Submission date:	14 September 2015
	Editorial Decision:	28 October 2015
	Revision received:	28 January 2016
	Editorial Decision:	04 March 2016
	Revision received:	10 March 2016
	Accepted:	14 March 2016

Editor: Barbara Pauly

Transaction Report:

1st Editorial Decision

28 October 2015

Thank you very much for the submission of your research manuscript to our editorial office. First of all I would like to apologize for the unusual amount of time it has taken us to complete the review process. We have now received the full set of reports from the referees that were asked to assess it. As the detailed reports are pasted below I will only repeat the main points here.

The referees point out a number of technical and experimental concerns that would need to be addressed. More importantly however, it also becomes clear from the reports and the additional communication that took place during our standard cross-commenting process in which we invite the referees to feedback on each others' reports, that all three referees feel that further insights into the mechanism by which folliculin affects lysosomal movements and mobility is necessary. As it stands and without these insights, the reviewers do not feel that the study is suitable for publication here.

From the analysis of these comments it becomes clear that significant revision is required before the manuscript becomes suitable for publication in EMBO reports. However, given the potential interest of your study on which all referees agreed, I would like to give you the opportunity to revise your manuscript, with the understanding that the main concerns of the referees must be addressed and their suggestions taken on board, also with regard to providing further insights into the role of folliculin in controlling lysosome dynamics. Acceptance of the manuscript will depend on a positive outcome of a second round of review and I should also remind you that it is EMBO reports policy to allow a single round of revision only and that, therefore, acceptance or rejection of the manuscript

will depend on the completeness of your responses included in the next, final version of the manuscript.

Revised manuscripts should be submitted within three months of a request for revision; they will otherwise be treated as new submissions. If you feel that this period is insufficient for a successful submission of your revised manuscript I can potentially extend this period slightly.

I look forward to seeing a revised form of your manuscript when it is ready. Should you decide to seek rapid publication somewhere else instead, I would appreciate a short note as well.

REFEREE REPORTS

Referee #1:

The manuscript 'Folliculin cooperates with Rab34 to control nutrient dependent changes in lysosome Positioning' describes novel factors that control the cellular localization of lysosomes, in this case under the control of starvation. The authors suggest that Folliculin and Rab34 control starvation induced relocalization of lysosomes, as well as perinuclear localization under basal conditions in some cells, probably by restricting endosomal motility. A similar point has been suggested for manipulation of cholesterol that then controls the binding of the dynein motor to RILP through the formation of ER contact sites (Rocha et al., 1999) and this point should be incorporated in the author's arguments. Especially because the authors demonstrate that Folliculin also binds to RILP and localizes to Rab34, both in a manner dependent on its DENN domain, and that starvation induces colocalization of Rab34 with the lysosomes. Given the localization of Rab34 to the golgi-network, the authors speculate that the Golgi forms contact sites with the lysosomes to restrict their motility.

The manuscript deals with a timely topic, intercompartmental regulation, and introduces an interesting concept. Overall, the experiments are well executed and address a highly interesting topic. I highly liked the manuscript. Yet, weaknesses of the paper include limited mechanistic insight and lack of further investigation into the existence of contact sites between the golgi and the lysosomes, the latter of which I feel is necessary to warrant publication in EMBO Reports.

Major comments:

- The authors propose a model in which the golgi restricts lysosomal motility by virtue of membrane contact sites. These contact sites should be directly visualized by EM for example by doing Immuno-EM for Rab34 and Lamp1). Furthermore, in Figure 5D the lysosomes (LysotrackerRed) appear to be inside the Rab34 vesicles, rather than in contact with them. Also, no golgi-marker staining is used together with Rab34 and the lysosomes (are the vesicles in the periphery that contain Rab34 still positive for golgi-markers?).

- The authors argue that the interaction should be between FLCN-RILP-Rab35 while an FLCN-RILP-Rab7 interaction is (to my opinion) not excluded. The authors should exclude this interaction but -I believe- showing by EM a direct contact site between late endosomes/lysosomes and golgi in a Rab35 dependent manner should do.

- FLCN localizes to lysosomes in a DENN-domain and FNIP dependent manner (Petit et al, 2013). Given the DENN dependent localization of FLCN to Rab34 and the failure of FNIP to localize to Rab34, how do the authors reconcile that FLCN induces contact site formation between the TGN and lysosomes? Which interaction is essential for the contact site, FLCN-Rab34 or FNIP-FLCN? Furthermore it would be interesting to know what the effect is of FNIP knockdown on lysosome relocalization, since the effects of FLCN can be FNIP dependent but Rab34 independent.

- The authors do not show that the starvation induced colocalization between Rab34 and the lysosomes is dependent on FLCN. Can they also explain why FLCN under starved conditions migrate differently by SDS-PAGE (Fig 1)?

Minor comments

- The DENN domain is 40% of the FLCN protein. Smaller deletions would be useful to distinguish

the effects of FLCN via Rab34 and via FNIP.

- The EGFR degradation experiment is not convincing and no statistics are provided.
- The manuscript is very elaborate and especially the discussion would benefit from some more compact writing. Furthermore, it misses a concluding paragraph with the message of the paper.
- The manuscript (especially the discussion) contains many incorrect sentences. For example: 'We also showed that by live-cell imaging that FLCN dynamically associates'.
- The > and < seem to be switched in the text for figure 1 (<85%), Figure 1B and in the text for figure 6(<100 fold).
- Fig 6a. The authors should not show overexposed lanes. Especially the lanes with Rab7 are overexposed and show (tmo) less Rab7 in the middle lane of the input sample. If so, relatively more Rab7 is obtained in the RILP pull-down.

Referee #2:

The protein folliculin (FLCN) has attracted considerable attention as mutations cause renal tumors, and it has been linked to control of mTOR signalling. It has been suggested to be a GAP that controls one of the GTPases that regulates mTOR. This paper provides evidence to argue that FLCN can also control the intracellular distribution of lysosomes and does so by interacting with Rab34, a GTPases previously shown to be involved in lysosomal localization via its effector RILP (a known dynein adaptor).

The data in the paper are of a good technical quality with excellent quality micrographs and some useful quantitation and controls. However the paper is rather confusing with a lot of text in every section but with no clear picture of mechanism emerging. Rab34 is known to be primarily on the Golgi, whilst FLCN is on lysosomes. The authors make a reasonable case that in some circumstances the proteins can co-localize. However the link between the proteins is unclear. Primarily it seems to be that when FLCN levels are altered, then the distribution of lysosomes is affected, and so they test Rab34 as one of the GTPases known to interact with RILP, a protein that controls lysosome distribution. They provide evidence that Rab34 can recruit FLCN directly to membranes (although not the normal binding partner of FLCN, FNIP2), and also that FLCN can bind directly to RILP. But how these interactions lead to increased lysosomal movement is unclear. The final section of re-expressing FLCN in a cell line that lacks the protein provides a good confirmation that FLCN can affect lysosomal distribution, but the link with Rab34 is again tenuous as the effect on levels of Rab34 are rather subtle (Figure 6A). Thus it seems clear that FLCN can affect lysosome distribution by a direct or indirect mechanism, and that Rab34 can in unusual circumstances interact with it, even though the proteins are mostly in different parts of the cell, but what this means in practise is not clear.

I was also concerned about the best evidence that there is a link between Rab34 and FLCN. The authors relocalize Rab34 to mitochondria and find that over-expressed GFP-tagged FLCN follows. However it is odd that FNIP2 (a binding partner for FLCN) does not relocate, and this raises the concern that the FLCN on mitochondria is not correctly folded. In addition, it is not clear how the Rab34 is activated on mitochondria as its exchange factor will not be there. Finally the Rab34-dsRed-Mito shows a very different distribution on the mitochondria to the FLCN it is supposed to be interacting with (most obvious in Figure 5D). This is of course not possible if the proteins are really binding, and it raises the concern that the mitochondria are damaged by the Rab construct and the over-expressed FLCN is binding non-specifically. In the end, if the authors believe that FLCN is a Rab34 effector, or a Rab34 GEF, they should be able to show this with direct biochemical assays, ideally with purified recombinant proteins.

Given these concerns I regret that, despite the really good imaging, the paper overall is insufficiently conclusive to be suitable for publication.

Referee #3:

As the authors have comprehensively detailed it has become apparent through recent work that lysosome positioning in response to nutrient signalling may play an important role in cellular homeostasis. In this manuscript the authors propose that FLCN, which is required for mTORC1

recruitment and activation, may also function by regulating membrane traffic. FLCN has also been shown to be involved in several other cellular processes some of which also connect it to the lysosome.

To support their proposed function of FLCN the authors provide data that show clustering of lysosomes in starvation requires FLCN and overexpression of FLCN and FNIP2 cause tubulation of lysosomes. They further show that Rab34, a Rab protein known to be involved in lysosome dynamics, also affects lysosome clustering, in a starvation-dependent manner similar to FLCN, and that it is likely the two proteins act in the same pathway. This data is supported by live cell imaging that shows FLCN and Rab34 are present of membranes which interact with lysosomes, and overexpression of Rab34 affects the mobility of the lysosomes. Importantly the authors confirm the localization with reagents to the endogenous Rab34 and FLCN.

Ectopic targeting of Rab34 showed that FLCN could be recruited to Rab34-positive mitochondria, suggesting they form a complex. As starvation activates Rab34 as measured by RILP pulldowns, they investigate the complex and its binding to RILP. Using constructs targeted to mitochondria, FLCN required its DENN domain to bind Rab34, and in pulldowns FLCN with its DENN domain, but not the N-terminus, could bind the RILP.

Finally the authors study the BHD kidney cancer cell line, UOK257, with and without FLCN. They show that FLCN is required for increased association of Rab34 with RILP, and confirm that lysosome dynamics is controlled by FLCN. This and other data from the OUK257 support the model that FLCN regulation of lysosomes involves Rab34/RILP.

The authors should be commended for the development of methods to quantify subcellular distributions of organelles which is particularly challenging, and for their comprehensive coverage to the literature, in particular many original references.

General comments

Overall the data are well done and clearly presented and while they provide evidence for localizations, perturbation of localization and interactions there is no mechanism for how the interaction between FLCN- Rab34 and RILP would control lysosome dynamics, nor is there any insight into how this would impact on the outcome of the altered response, for example mTORC1 signalling, TFEB activation or other responses downstream of altered nutrient levels. Is phosphorylation of FLCN involved in the complex formation or stability? Regarding the FLCN-RAB34-RILP complex, how would this assemble and can they demonstrate a direct interaction between the 3 proteins? and can the authors show any regulation of the complex by activation of Rab34, or starvation? What happens to the complex when nutrients are restored and lysosomes revert to their original state? What membranes, if any, are involved besides lysosomes? In light of these open questions the data is largely descriptive.

Specific comments

1. Page 13. How did the authors match Rab34 expression level in transfected cells?
2. Page 14 Figure 4A, how do the authors know the GFP-Rab34 was associated with the Golgi?
3. Some parts of the text is confusing. For example see page 16 5th line from bottom "We designed ..." a scheme of the constructs would be useful.
4. It was not clear why the EGF receptor data was included. Is the difference statistically significant? Aside from motility and localization of lysosomes what specific mechanism is behind the FLCN-dependent increase in degradation? Increased fusion with lysosomes? altered pH? can they exclude altered receptor dynamics?
5. Page 23, what are the "heterologous" Rab34 positive membranes?

1st Revision - authors' response

28 January 2016

Referee #1: The manuscript 'Folliculin cooperates with Rab34 to control nutrient dependent changes in lysosome Positioning' describes novel factors that control the cellular localization of lysosomes, in this case under the control of starvation. The authors suggest that Folliculin and Rab34 control

starvation induced relocalization of lysosomes, as well as perinuclear localization under basal conditions in some cells, probably by restricting endosomal motility. A similar point has been suggested for manipulation of cholesterol that then controls the binding of the dynein motor to RILP through the formation of ER contact sites (Rocha et al., 1999) and this point should be incorporated in the author's arguments. Especially because the authors demonstrate that Folliculin also binds to RILP and localizes to Rab34, both in a manner dependent on its DENN domain, and that starvation induces colocalization of Rab34 with the lysosomes. Given the localization of Rab34 to the golgi-network, the authors speculate that the Golgi forms contact sites with the lysosomes to restrict their motility. The manuscript deals with a timely topic, intercompartmental regulation, and introduces an interesting concept. Overall, the experiments are well executed and address a highly interesting topic. I highly liked the manuscript. Yet, weaknesses of the paper include limited mechanistic insight and lack of further investigation into the existence of contact sites between the golgi and the lysosomes, the latter of which I feel is necessary to warrant publication in EMBO Reports.

We appreciate the reviewer's enthusiasm for our paper. We have performed a series of biochemical experiments to address the reviewer's concerns regarding mechanism (discussed at length in response to reviewers 2 and 3). We have also further explored the contact sites using a series of imaging techniques. We have referenced the Rocha *et al.* 2009 paper in the discussion (pg 21).

Major comments:

The authors propose a model in which the golgi restricts lysosomal motility by virtue of membrane contact sites. These contact sites should be directly visualized by EM for example by doing Immuno-EM for Rab34 and Lamp1). Furthermore, in Figure 5D the lysosomes (LysotrackerRed) appear to be inside the Rab34 vesicles, rather than in contact with them. Also, no golgi-marker staining is used together with Rab34 and the lysosomes (are the vesicles in the periphery that contain Rab34 still positive for golgi-markers?).

The authors argue that the interaction should be between FLCN-RILP-Rab34 while an FLCN-RILP-Rab7 interaction is (to my opinion) not excluded. The authors should exclude this interaction but -I believe-showing by EM a direct contact site between late endosomes/lysosomes and golgi in a Rab34 dependent manner should do.

Taking the two points above together, we have performed a series of super-resolution and ultrastructural imaging experiments to better define the nature of Rab34-lysosome contacts that are presented in figure EV5. Using 160x deconvolution microscopy, SIM-super resolution and both conventional and cryo-immuno electron microscopy, we believe that we now provide convincing evidence of the existence of these contacts (pg 13).

To demonstrate that Rab34 localizes to the Golgi under our conditions, we have supplied some additional immunofluorescence images in Figure EV3, co-staining GFP-Rab34 transfected cells with either GM130 (cis Golgi), Giantin (cis/medial Golgi) or Golgin-97 (mainly TGN). There is a good degree of overlap of Rab34 with all 3 Golgi markers (pg 1112). This demonstrates that GFP-Rab34 is predominantly Golgi-localised although we cannot rule out some low levels in other compartments. We decided to retain our description of Rab34 localization to a conservative 'perinuclear membranes' because of this. It is not straightforward to do this with endogenous proteins as we use mouse and rabbit antibodies for LAMP1/Rab34. On the point about the lysosome being inside a Rab34 positive vesicle, at this resolution it is not possible to exclude this possibility, but we believe that the Rab34 compartment partially wraps around the lysosome. We chose this example as a contact that could be spatially resolved compared to the more dense ones nearer the nucleus and because the movement of both signals together is striking – we make this clear in the text (pg 13) and supplementary movie 3 shows a whole cell and so the reader should be able to place this in its appropriate context.

With respect to Rab7, our new biochemical results in Figure 5 (and discussed on pg 16-17), showing that FLCN-DENN promotes association of Rab34 and RILP *in vitro*, clearly ties these three proteins together functionally and is consistent with the contact site model. The model we present still requires RILP to have capacity to associate with lysosomes, and Rab7 would be the obvious candidate to support this. Similarly, lysosomes presumably still have to move towards the minus end of microtubules, in part, through a Rab7-RILP-dynein axis. This is now mentioned in the discussion (pg 21).

FLCN localizes to lysosomes in a DENN-domain and FNIP dependent manner (Petit et al, 2013). Given the DENN dependent localization of FLCN to Rab34 and the failure of FNIP to localize to Rab34, how do the authors reconcile that FLCN induces contact site formation between the TGN and lysosomes? Which interaction is essential for the contact site, FLCN-Rab34 or FNIP-FLCN? Furthermore it would be interesting to know what the effect is of FNIP knockdown on lysosome relocalization, since the effects of FLCN can be FNIP dependent but Rab34 independent.

We have provided additional data (in Figure EV2) with regard to the lack of FNIP localization to mitochondria and discussed our interpretation of the results in our response to referee 2. There is considerably less FNIP2 than FLCN, and FNIP2 has an intrinsic capacity to localize strongly lysosomes (in a nutrient independent manner, Figure EV2, C,D), and the FLCN-DENN interaction with Rab34 is most likely not strong enough to pull it away from lysosomes. The FLCN we observe associated with mitochondria represents the large excess of the single protein. We have discussed this in the main text (pg 15). We use this method as a cell based, equilibrium binding assay, and we have added text that at the beginning of this section that makes this more explicit (pg 14).

At the reviewer's request, we have performed siRNA knockdown experiments targeting FNIP1/2 (Figure EV2A, B). We confirm that FNIP1/2 depletion prevents FLCN association with lysosomes and inhibits starvation dependent clustering. We conclude that formation of the FNIP-FLCN complex is essential for this function of FLCN – most likely in promoting FLCN recruitment to lysosomes (pg 8-9). In our hands, the FNIP antibodies are not good enough for western blotting or immunofluorescence analysis so we have not been able to analyse an endogenous complex of FNIP-FLCN.

The authors do not show that the starvation induced colocalization between Rab34 and the lysosomes is dependent on FLCN.

We have added data addressing this point in Figure EV4B: we depleted FLCN and examined the localisation of Rab34. Given there are fewer lysosomes in the peri-nuclear region following FLCN depletion, there will inevitably be less co-localisation/contacts between Rab34 and lysosomes, and thus quantification of colocalization is difficult. However, we noticed fewer bright puncta of Rab34 in and around the Golgi. The panels show some representative images.

Can they also explain why FLCN under starved conditions migrate differently by SDS-PAGE (Fig 1)?

Several studies have shown that FLCN is a phosphoprotein, and that suppression of mTORC1/AMPK activity modulates this phosphorylation state and affects electrophoretic mobility. We have amended the text with a line on pg 7 that reads “also resulted in a slight increase in the electrophoretic mobility of FLCN that is likely to occur as a result of a change in phosphorylation status (Figure 1B)” and referenced Baba *et al.* 2006 which provides some good examples of this.

Minor comments The DENN domain is 40% of the FLCN protein. Smaller deletions would be useful to distinguish the effects of FLCN via Rab34 and via FNIP.

The DENN domain is indeed a large proportion of the protein and more refinement would be desirable. However, it also possesses a complex, evolutionarily conserved fold (Nookala et al (2012), PDB:3V42). and thus it is not possible to make any deletions within this domain that one

could reasonably assume would still result in a properly folded protein. The question about FNIP binding is an interesting one. Whilst one study has implicated this domain in FNIP binding (Baba *et al*, 2006), the yeast FLCN homologue Lst7 (which lacks the DENN domain) and Lst4 (FNIP) are still capable of forming a complex (Pacitto *et al* 2015, Péli-Gulli *et al* 2015). This suggests that FNIP binding determinants are more complex and not likely to be resolved by a straightforward deletion strategy. .

The EGFR degradation experiment is not convincing and no statistics are provided.

As noted in our response to reviewer 3 (who questioned whether this really belonged in the paper at all), we have removed these data, since they are only very peripheral to our conclusions.

The manuscript is very elaborate and especially the discussion would benefit from some more compact writing. Furthermore, it misses a concluding paragraph with the message of the paper. -The manuscript (especially the discussion) contains many incorrect sentences. For example: 'We also showed that by live-cell imaging that FLCN dynamically associates'. -The > and < seem to be switched in the text for figure 1 (<85%), Figure 1B and in the text for figure 6(<100 fold).

We have done our best to correct grammatical errors. We have extensively revised and shortened all sections of the manuscript and particularly the discussion.

Fig 6a. The authors should not show overexposed lanes. Especially the lanes with Rab7 are overexposed and show (tmo) less Rab7 in the middle lane of the input sample. If so, relatively more Rab7 is obtained in the RILP pull-down.

The Rab7 input blot was within the linear range of the image quantification system we use, but we agree it did appear over-exposed in the figure. We have replaced it with a shorter exposure. Rab7 levels do appear slightly lower in this blot – but overall, we did not observe a difference in Rab7 expression between these two cell lines nor in multiple repeats did we see a significant difference in apparent activity.

Referee #2: The protein folliculin (FLCN) has attracted considerable attention as mutations cause renal tumors, and it has been linked to control of mTOR signalling. It has been suggested to be a GAP that controls one of the GTPases that regulates mTOR. This paper provides evidence to argue that FLCN can also control the intracellular distribution of lysosomes and does so by interacting with Rab34, a GTPases previously shown to be involved in lysosomal localization via its effector RILP (a known dynein adaptor). The data in the paper are of a good technical quality with excellent quality micrographs and some useful quantitation and controls. However the paper is rather confusing with a lot of text in every section but with no clear picture of mechanism emerging. Rab34 is known to be primarily on the Golgi, whilst FLCN is on lysosomes. The authors make a reasonable case that in some circumstances the proteins can co-localize. However the link between the proteins is unclear. Primarily it seems to be that when FLCN levels are altered, then the distribution of lysosomes is affected, and so they test Rab34 as one of the GTPases known to interact with RILP, a protein that controls lysosome distribution. They provide evidence that Rab34 can recruit FLCN directly to membranes (although not the normal binding partner of FLCN, FNIP2), and also that FLCN can bind directly to RILP. But how these interactions lead to increased lysosomal movement is unclear. The final section of re-expressing FLCN in a cell line that lacks the protein provides a good confirmation that FLCN can affect lysosomal distribution, but the link with Rab34 is again tenuous as the effect on levels of Rab34 are rather subtle (Figure 6A). Thus it seems clear that FLCN can affect lysosome distribution by a direct or indirect mechanism, and that Rab34 can in unusual circumstances interact with it, even though the proteins are mostly in different parts of the cell, but what this means in practise is not clear.

We appreciate the reviewer's comments. We have shortened and simplified the paper, improved our data connecting FLCN and Rab34 and RILP directly as described below, and thereby provide some

new key insights into molecular mechanism. Some of the effects on apparent activity/expression are not large, but they are reproducible, statistically significant and within the range expected of this type of effector pull down assay (see work from the Dikic lab – McEwan *et al.* 2015, Cell Host and Microbe Figure S2H and Braga lab – Carroll *et al.* Dev Cell 2013 Figure 3C for examples).

I was also concerned about the best evidence that there is a link between Rab34 and FLCN. The authors relocate Rab34 to mitochondria and find that over-expressed GFP-tagged FLCN follows. However it is odd that FNIP2 (a binding partner for FLCN) does not relocate, and this raises the concern that the FLCN on mitochondria is not correctly folded.

We agree that the interpretation of this experiment is complex. In general, ectopic FLCN is expressed at a much higher level than FNIP2, whether transfected singly or together. Indeed, Tsun *et al.* (2013, Sabatini lab) noted that “*For FLCN-FNIP purifications, it is crucial to immunoprecipitate through the FNIP component to obtain stoichiometric complexes with high activities*”. This strongly suggests that under these conditions, there is a free pool of (non-FNIP bound) FLCN. We propose that this free FLCN is what is interacting with Rab34 on mitochondria. Moreover, FNIP2 localises almost entirely to lysosomes with very little cytoplasmic background following over expression in a nutrient-independent manner, and will take both endogenous and over-expressed FLCN (Fig 2, Figure EV2C) with it. FNIP2 is most likely not recruited to mitochondria because it has a much higher affinity for lysosomes and there is much less of it than FLCN. We have amended the text (pg 15) and provided supplementary data in Figure EV2 to support this proposition:

“Following transfection, FLCN tends to be expressed at a much higher level than FNIP proteins (Figure EV2D), so given the nutrient-independent strong localization of FNIP2 to lysosomes (Figure EV2C), it is likely that the mitochondrial-bound FLCN represents a non-FNIP bound cytoplasmic pool”

It is possible that full length FLCN is not properly folded, although it shows no sign of aggregation or degradation. However it is harder to make that critique for the DENN domain alone, which we also show localises to Rab34-mitochondria. This construct is essentially identical to that used by Nookala *et al.* (2012) for crystallographic studies (expressed in *E. coli*), and in our hands expresses with high yield and is very stable. It is unlikely that this would be mis-folded in a mammalian expression system.

In addition, it is not clear how the Rab34 is activated on mitochondria as its exchange factor will not be there. Finally the Rab34-dsRed-Mito shows a very different distribution on the mitochondria to the FLCN it is supposed to be interacting with (most obvious in Figure 5D). This is of course not possible if the proteins are really binding, and it raises the concern that the mitochondria are damaged by the Rab construct and the over-expressed FLCN is binding non-specifically.

The reviewer questions the validity of this ectopic targeting assay. Mitochondrial targeting of proteins has been used successfully on a number of occasions to relocate binding partners (we reference Bear *et al.* 2000). Another example is shown in ‘knock sideways’ experiments by Robinson *et al.* Dev Cell 2010: *Rapid Inactivation of Proteins by Rapamycin-Induced Rerouting to Mitochondria*. Whilst we agree one must be cautious in interpretation of the results, we do show that there is specificity here by the fact that Rab35 does not induce FLCN recruitment to mitochondria. Figure 5D (now EV6D) is a super resolution image – the reason why we use this approach is to spatially resolve these two signals and demonstrate that FLCN is targeted to the mitochondrial surface, not to its interior. The activation status of the Rab, we agree is not entirely unambiguous, although one would expect the Q mutant to be GTP bound as, given the large (≈ 10 fold) excess of GTP over GDP in the cell (Traut 1994 Molecular and Cellular Biochemistry v140 pg 1-22 and Pitz *et al.* 1997 Cell Growth and Differentiation v8 pg 53-59) it would be expected to bind GTP as it comes off the ribosome, and not hydrolyse it. A GEF is therefore not required. We present this data simply as evidence that Rab34 is capable of interacting with FLCN, at equilibrium, in a cellular context and we don’t think we over-interpret it. However, we have modified the text to reflect the caution that the reader should use in interpretation of these data – it now reads (pg 15)

“Transfection of the *Q111L* mutant resulted in recruitment indistinguishable to that observed for wildtype *Rab34* suggesting that *FLCN* may interact with *Rab34* in the active state (Figure EV6C, bottom). However, the true nucleotide state of these ectopically targeted GTPases is ambiguous and we were unable to detect any expression of the *T66N-Mito* fusion protein despite deriving and sequencing multiple constructs.”. Now reinforced by the direct biochemical analysis requested by the reviewer described below, we think these data are much stronger.

In the end, if the authors believe that *FLCN* is a *Rab34* effector, or a *Rab34* GEF, they should be able to show this with direct biochemical assays, ideally with purified recombinant proteins.

At the reviewer’s request, we have explored this and now provide a crucial new mechanistic insight that we believe should assuage their concerns (pg 16-17). We have focused on the *FLCN-DENN* domain that is soluble, folded and stable (using the same amino acids as in the crystallographic study of Nookala *et al.* 2012). We have exhaustively confirmed previous observations using purified recombinant proteins that the *FLCN-DENN* domain does not act as a *Rab34* GEF (Figure EV7), but rather, interacts directly with *RILP* and acts specifically to load it with active, GTP bound, *Rab34* (Figure 5). We are very excited by these new data, which we believe significantly improve our study by linking *FLCN*, *Rab34* and *RILP* together via a mechanism that is consistent with our cellular assays and model. We are grateful to the reviewers for their thoughtful comments that have led to this new data. Some additional comments on this are made in our response to reviewer 3.

Given these concerns I regret that, despite the really good imaging, the paper overall is insufficiently conclusive to be suitable for publication.

In summary, we have provided new biochemical analysis using purified recombinant proteins that the reviewer has requested that now provides a key insight into the molecular mechanism by which *FLCN* *directly* controls lysosome distribution through *Rab34* and *RILP*. We think that this is much more conclusive and we hope that the reviewer will reconsider our manuscript.

Referee #3: As the authors have comprehensively detailed it has become apparent through recent work that lysosome positioning in response to nutrient signalling may play an important role in cellular homeostasis. In this manuscript the authors propose that *FLCN*, which is required for *mTORC1* recruitment and activation, may also function by regulating membrane traffic. *FLCN* has also been shown to be involved in several other cellular processes some of which also connect it to the lysosome. To support their proposed function of *FLCN* the authors provide data that show clustering of lysosomes in starvation requires *FLCN* and overexpression of *FLCN* and *FNIP2* cause tubulation of lysosomes. They further show that *Rab34*, a Rab protein known to be involved in lysosome dynamics, also affects lysosome clustering, in a starvation-dependent manner similar to *FLCN*, and that it is likely the two proteins act in the same pathway. This data is supported by live cell imaging that shows *FLCN* and *Rab34* are present of membranes which interact with lysosomes, and overexpression of *Rab34* affects the mobility of the lysosomes. Importantly the authors confirm the localization with reagents to the endogenous *Rab34* and *FLCN*. Ectopic targeting of *Rab34* showed that *FLCN* could be recruited to *Rab34*-positive mitochondria, suggesting they form a complex. As starvation activates *Rab34* as measured by *RILP* pulldowns, they investigate the complex and its binding to *RILP*. Using constructs targeted to mitochondria, *FLCN* required its *DENN* domain to bind *Rab34*, and in pulldowns *FLCN* with its *DENN* domain, but not the N-terminus, could bind the *RILP*. Finally the authors study the BHD kidney cancer cell line, *UOK257*, with and without *FLCN*. They show that *FLCN* is required for increased association of *Rab34* with *RILP*, and confirm that lysosome dynamics is controlled by *FLCN*. This and other data from the *UOK257* support the model that *FLCN* regulation of lysosomes involves *Rab34/RILP*. The authors should be commended for the development of methods to quantify subcellular distributions of organelles which is particularly challenging, and for their comprehensive coverage to the literature, in particular many original references. General comments Overall the data are well done and clearly presented and while they provide evidence for localizations, perturbation of localization and

interactions there is no mechanism for how the interaction between FLCN-Rab34 and RILP would control lysosome dynamics,

We are grateful for the reviewer's overall opinion of our manuscript. We have now provided what we believe is a key insight into the molecular mechanism linking FLCN, RILP and Rab34 to lysosome dynamics: *in vitro* the FLCN-DENN domain acts to load active Rab34 onto RILP (see details below). We think that this mechanism is interesting and novel as it is the first time (to our knowledge) that a DENN domain protein has been shown to behave in such a way.

nor is there any insight into how this would impact on the outcome of the altered response, for example mTORC1 signalling, TFEB activation or other responses downstream of altered nutrient levels.

We accept that here we have not explored these downstream effects on the pathways that FLCN has been implicated in such as mTORC1, autophagy, TFEB (which is most likely downstream of mTORC1), ciliogenesis, B-cell maturation, cell cycle, cell adhesion/migration etc. The reason for this is that many of these phenotypes are not strong, are still controversial within the field, and are highly context dependent. What we provide here is a conceptual advance, which we believe is important *in its own right*, by expanding our knowledge of the mechanisms that control the spatial organisation of lysosomes, which should provide new perspective in understanding these other functions of FLCN, and could explain some inconsistencies within the field.

Is phosphorylation of FLCN involved in the complex formation or stability?

The shift in the electrophoretic mobility of FLCN following starvation suggests that FLCN association with lysosomes is in part controlled by phosphatase/kinase activity. This is certainly worth investigating in the future. However, FLCN appears to have a minimum of 5 phosphorylation sites (S62, S302, S406, S537, S542) (Wang *et al.* 2010 FEBS letters, Dunlop *et al.* Autophagy 2015, Piao *et al.* 2009 BBRC) Web tools such as Scansite and Phosphosite predict several more. There are a minimum of 3 kinases implicated (mTORC1, AMPK, and ULK1) in FLCN phosphorylation and potential feedback loops to the same kinases. The FNIPs are also phosphoproteins (Baba *et al.* 2006). No phosphatases have been identified. Pursuing this would be a big project in itself and we think that it is beyond the scope of what we can reasonably do here.

Regarding the FLCN-RAB34-RILP complex, how would this assemble and can they demonstrate a direct interaction between the 3 proteins?

At the reviewer's request, we have explored this and now offer a crucial new mechanistic insight that we hope should assuage their concerns (pg 16-17). We have focused on the FLCN DENN domain that is soluble, folded and stable (using the same amino acids as in the crystallographic study of Nookala *et al.* 2012). We confirm using purified recombinant proteins that the FLCN-DENN domain does not act as a Rab34 GEF (Figure EV7), but instead interacts directly with RILP and acts specifically to load it with only active, GTP bound Rab34 (Figure 5). The fact that Rab34 binding appears to saturate yet this effect is observed with a sub-stoichiometric amount of FLCN compared to Rab34 in the pull down suggests that this does not require the formation of a stable ternary complex. We are very excited by these new data, which we believe significantly improves our study and elegantly ties FLCN, Rab34 and RILP together, with a mechanism that is consistent with our model. In terms of how such a complex assembles, structural analysis provides some clues. If one assumes that the FLCN DENN domain interacts with Rab34 in a similar manner to DENN1B interaction with Rab35 (PDB:3TW8) and that Rab34 interacts with RILP in a similar manner to Rab7 (PDB:1YHN), then following some homology modelling and alignments it becomes clear that the formation of a (perhaps transient) tri-partite complex is very plausible. FLCN, with the capacity to interact with both components, could regulate their association and/or dissociation kinetics. We think this is too speculative to include in the manuscript, but we will use the homology model generated below using a combination of PHYRE2 one-2-one threading (Rab34 into Rab7 in

PDB:1YHN) and Pymol 'align' functions (FLCN DENN (PDB:3V42) into Rab35-DENN1B, and finally Rab35 into Rab34

and can the authors show any regulation of the complex by activation of Rab34, or starvation? What happens to the complex when nutrients are restored and lysosomes revert to their original state? What membranes, if any, are involved besides lysosomes? In light of these open questions the data is largely descriptive.

We have provided the direct biochemical analysis requested by the reviewer that now provides key new insight into the molecular mechanism by which FLCN *directly* controls lysosome distribution through RILP and Rab34. We think that this elevates our study well above being descriptive. There are still of course many open questions, but we believe that we have provided a new function for a heavily studied but still enigmatic protein, coupling the key cellular processes of nutrient signalling and lysosome dynamics, where we have identified the central players and now offer a coherent molecular mechanism and model.

Specific comments

1. Page 13. How did the authors match Rab34 expression level in transfected cells?

Cells were selected for quantification that expressed Rab34 robustly (similar levels to our IF) but not at an excessively high level. In our system, these cells had mean raw fluorescence intensities (over the whole cell) from the same exposure time in the range of 3000-10000 grey values (in 16-bit images). We confirmed that we had selected a comparable population for each oligonucleotide by comparing the mean of these values in each case. We have added this information to the methods (pg 42) and also provide an additional graph in Figure 3D with these data.

2. Page 14 Figure 4A, how do the authors know the GFP-Rab34 was associated with the Golgi? We have supplied some additional immunofluorescence images (Figure EV3), co-staining GFP-Rab34 transfected cells with either GM130 (cis Golgi), Giantin (cis/medial Golgi) or Golgin-97 (mainly TGN). There is a good degree of overlap with all 3 markers. This shows that most GFP-Rab34 is Golgi-localised although we cannot rule out that a small proportion localises on other compartments. We keep our description to a conservative 'peri-nuclear membranes' because of this.

3. Some parts of the text is confusing. For example see page 16 5th line from bottom "We designed ..." a scheme of the constructs would be useful. We have clarified the text here and have extensively revised the whole manuscript. We have also included a schematic of the mitochondrial targeting construct in Figure EV6.

4. It was not clear why the EGF receptor data was included. Is the difference statistically significant? Aside from motility and localization of lysosomes what specific mechanism is behind the FLCN-dependent increase in degradation? Increased fusion with lysosomes? altered pH? can they exclude altered receptor dynamics?

We performed this experiment to begin to examine whether the difference in lysosome dynamics had the potential to impact upon on this well characterized lysosomal function. However, we take

the reviewer's point that this is overly simplistic and it is clear that the reasons for this difference in degradation may be much more complex than that just distribution/dynamics, and would require a lot of additional validation. As these data are only very peripheral to our study, we have removed them.

5. Page 23, what are the "heterologous" Rab34 positive membranes?

We have removed this phrase.

2nd Editorial Decision

04 March 2016

Many thanks for your patience while we were assessing the revised version of your manuscript. I have now received feedback from the referees on your revision.

Based on these discussions, I am happy to tell you that we would like to offer publication of the study in EMBO reports. Referee 2 still raises two points that s/he would like to have clarified, but upon further discussions with the other reviewers it would be sufficient if you experimentally addressed the following point:

Referee 2 is not convinced that simultaneous binding of RILP to Rab7 and Rab34 is possible. Therefore, I would like to ask you to add one experiment, in which you express tagged Rab7, Rab34 and RILP and show that by isolating Rab7, RILP and Rab34 are co-isolated and that this requires RILP expression. This experiment could also be done by isolating Rab34 and then testing co-IP of RILP and Rab7.

There are also a few formal things that I would like you to address before acceptance.

1. Currently, the manuscript is submitted as a Scientific Report. As such, it correctly has 5 figures, but it should have a combined results & discussion section, which is currently not the case. In contrast, our full articles have a separate results and discussion section, but should have more than five figures. It is up to you, but the easiest thing to do would be to move one or ideally two of the supplementary figures into the main manuscript and convert it to a full article. This way you don't have to combine the R&D section and the fact that the text is a little longer than our usual Scientific Reports would also not matter anymore.

2. Please also update the references to the supplementary movies (they should be called and referenced as "MovieEV1" etc.)

3. Please note that the order of panels in Fig 5 seems to be out of order (panel F is before panel E).

4. Please also provide us with a conflict-of-interest statement and with an author contribution statement. For more information, please visit our website.

5. I would also like to point out that we now strongly encourage the publication of original source data (whole western blots and raw microscopical images) with the aim of making primary data more accessible and transparent to the reader. The source data will be published in a separate source data file online along with the accepted manuscript and will be linked to the relevant figure. This is optional at the moment, but strongly encouraged.

Once you have addressed these points, please submit the final version of the study through our website again so that we can proceed with its official acceptance and rapid publication.

Again, many thanks for your contribution to our journal and with best wishes for a good weekend.

REFEREE REPORTS

Referee #1:

This is an interesting manuscript with beautiful images and extensive experiments. The authors make an interesting point; regulation of lysosomal positioning by a Golgi-associated GTPase, Rab34. This would add to the role of the ER in these processes through ER-late endosomal contact sites. Here, Golgi-endosomal contact sites are also postulated, then generated in response to nutrient depletion conditions.

Only small point: there are quite a number of misspellings in the Discussion. Please correct.

Referee #2:

This is a revised version of a paper that provides evidence to argue that the protein folliculin (FLCN) can control the intracellular distribution of lysosomes and does so by interacting with Rab34, a GTPase previously shown to be involved in lysosomal localization via its effector RILP (a known dynein adaptor). FLCN has attracted attention as a tumor-suppressor and has previously been proposed to act as a GAP for some of the GTPases that control mTOR signalling.

In revising this paper the authors have added new data to strengthen their argument that FLCN promotes an interaction between GTP-bound Rab34 and RILP and that in so doing it can direct the formation of links between the Golgi and lysosomes. They have also done a good job of shortening the text so that it is more succinct.

However the issue of mechanism is still rather unclear. The authors seem to be proposing that FLCN promotes binding between Rab34 on the Golgi (or "Rab34-positive perinuclear membranes") and RILP on lysosomes where it is presumably binding Rab7. This means that RILP now attaches lysosomes to the Golgi and so generates an "interorganelle contact site" that causes the lysosomes to cluster in the center of the cell. The problem with this model is firstly that the Hong lab has shown that the Rab7 and Rab34 binding sites on RILP are close to each other and so it seems hard to see how RILP could serve as a bridge between two compartments. More seriously, the Hong lab also reported that RILP-dependent regulation of lysosomes by Rab34 is insensitive to treating cells with brefeldin A to disperse the Golgi, indicating that the action of Rab34 is unlikely to be to form a bridge between Golgi and lysosomes but rather that it acts directly on lysosomes themselves (see Wang and Hong 2002).

Another concern I raised was that in the experiments where Rab34 is relocated to mitochondria, the FLCN-GFP that is now seen on mitochondria does not co-localize with the Rab34 which is inconsistent with the authors conclusion that the two proteins bind directly. In rebutting this the authors state that I am doubting the mitochondrial relocation assay and cite other papers using it. I do not doubt that the assay has potential value, but rather I am questioning the conclusion the authors which to make based on their data. When two proteins are clearly separated by at the resolution of light microscopy they cannot be binding directly to each other, and this is clearly the case even in the revised version of the paper (eg Figure 5B, DENN only panels).

Given this concern, and the lack of a clear mechanism emerging that can be summarized in a simple figure, then I suspect that despite the improvements of other aspects of the work, and the high technical quality of the imaging, the paper would confuse even a specialist reader and so it seems still unsuited for a general interest journal.

Referee #3:

The authors has addressed the most critical aspect of the my comments by providing the evidence that the FLCN DENN domain functions to load active Rab34 on to RILP. This provides an important insight into how FLCN controls the nutrient dependent distribution of lysosomes.

2nd Revision - authors' response

10 March 2016

We appreciate the opportunity to address the final comments of the reviewers.

We have performed the requested experiment in which we express GFP-Rab34 and HA-Rab7 and show that HA-RILP promotes their association. This is now shown in Figure 6I and described in the text on pg 18-19. As the reviewer says, Rab7 and Rab34 binding sites on RILP likely do overlap (Wang and Hong 2002), however RILP can also form a dimer (Wu et al. 2005 and Colucci et al. 2005) and so it is very plausible that it can interact with both GTPases simultaneously. We think that this is a useful additional to our paper. We have cited the additional papers and mentioned this in the discussion.

Although not specifically requested, we would also like to comment on the point on the BFA experiment. Although BFA does clearly disrupt the Golgi, it also has a wide range of effects on the ER as well as the morphology of endolysosomal system, as classic work by Jennifer Lippincott-Schwartz (Lippincott-Schwartz et al. Cell, 1991. Vol 67, pg 601) elegantly shows. Consequently, the reviewer's interpretation of the Wang and Hong experiment is not as straightforward as is indicated.

We have addressed your additional points in this revised manuscript and as suggested, increased the number of figures to article format, which we believe improves the presentation of the data.

As it is not yet mandatory, we have chosen not to include raw images and blots. The reason for this is that we almost always show images and movies of whole cells, in addition to zoom regions, and only ever make linear contrast adjustments. Similarly, we show whole membranes when appropriate and never aggressively crop or stitch together blots.

I would like to offer my thanks for a very constructive editorial and review process.

3rd Editorial Decision

14 March 2016

I am very pleased to accept your manuscript for publication in the next available issue of EMBO reports. Thank you for your contribution to our journal.

Corresponding Author Name: Mark Dodding

Journal Submitted to: EMBO REPORTS

Manuscript Number: EMBOR-2015-41382V2